# MetaFlowTrain: a highly parallelized and modular fluidic system for studying exometabolite-mediated inter-organismal interactions

Guillaume Chesneau[1], Johannes Herpell [1], Sarah Marie Wolf [1], Silvina Perin[1] & Stéphane Hacquard [1,2] ✉

Metabolic fluxes between cells, organisms, or communities drive ecosystem assembly and functioning and explain higher-level biological organization. Exometabolite-mediated inter-organismal interactions, however, remain poorly described due to technical challenges in measuring these interactions. Here, we present MetaFlowTrain, an easy-to-assemble, cheap, semi-high-throughput, and modular fluidic system in which multiple media can be flushed at adjustable flow rates into gnotobiotic microchambers accommodating diverse micro-organisms, ranging from bacteria to small eukaryotes. These microchambers can be used alone or connected in series to create microchamber trains within which metabolites, but not organisms, directionally travel between microchambers to modulate organismal growth. Using Meta-FlowTrain, we uncover soil conditioning effects on synthetic community structure and plant growth, and reveal microbial antagonism mediated by exometabolite production. Our study highlights MetaFlowTrain as a versatile system for investigating plant-microbe-microbe metabolic interactions. We also discuss the system´s potential to discover metabolites that function as signaling molecules, drugs, or antimicrobials across various systems.

Terrestrial and marine ecosystems are colonized by multi-kingdom microbial communities that are well-described in terms of composition and diversity[1-4]. However, the mechanisms underlying interactions between hosts, microbes, and the environment remain poorly understood. These interactions rely on direct contact between organisms but also metabolic exchanges between them that drive higher-level biological organization and co-existence or co-exclusion between organisms across various biological systems[5,6].

To date, researchers have often examined host–microbe and/or microbe–microbe interactions by assembling and mixing organisms in single-compartment gnotobiotic systems[7-10]. While these methods have been pivotal in deciphering interaction outputs and associated

mechanisms[11,12], they remain suboptimal for providing detailed insights into the metabolic fluxes and exchanges occurring between organisms or communities. Various strategies have been used to study exometabolite-mediated interactions between organisms, including halo assays[13], supernatant assays[14,15], and more complex systems like dual-chamber setups with porous membranes[16,17], porous microplates[18], modular microfluidic devices[19] or systems that include hosts[20,21]; however, these methods often fall short in one or more key areas. Some cannot track dynamic changes in microbial communities or exudate production over time, an essential capability since temporal sampling can produce vastly different insights[22]. Others are limited by their inability to accommodate a

[1]Department of Plant Microbe Interactions, Max Planck Institute for Plant Breeding Research, Cologne, Germany. [2]Cluster of Excellence on Plant Sciences (CEPLAS), Max Planck Institute for Plant Breeding Research, Cologne, Germany. ✉e-mail: hacquard@mpipz.mpg.de

broad range of organisms, lack high-throughput capabilities, or organismal compartmentalization. Additionally, many of these methods cannot collect samples non-destructively over time and rely on batch cultures, leading to the accumulation of specialized metabolites and nutrient depletion, and thus ultimately disrupting the natural dynamics of organism–to–organism communication[23].

To address these limitations, here we develop MetaFlowTrain, a fluidic system that offers several key advantages over current methods for the study of exometabolite-mediated interactions. (i) MetaFlowTrain accommodates a wide range of phylogenetically diverse organisms, including bacteria, fungi, microalgae, and multi-kingdom synthetic microbial communities (SynComs). (ii) The system is easily expandable and scalable, making it semi-high-throughput. (iii) MetaFlowTrain compartmentalizes microbial cells in gnotobiotic microchambers, facilitating exometabolic exchanges but not migration of microbial cells, allowing for precise study of metabolic fluxes between organisms. (iv) The system enables non-destructive, real-time metabolite collection over time. (v) Furthermore, it provides a fully adjustable nutrient flux that continuously sustains microbial growth, representing a major improvement over traditional batch culture.

In this paper, we demonstrate the MetaFlowTrain system's potential by highlighting its application in studying plant–microbe–microbe metabolic interplay. However, its versatility means that it can potentially be a valuable tool for addressing a broad range of research questions. We therefore discuss its wider applications and address its limitations, with a particular focus on studying inter-organismal communication.

## Results
### MetaFlowTrain is a highly parallelized and modular fluidic system
In the MetaFlowTrain (Fig. 1, Supplementary Movie 1, Supplementary Fig. 1) microorganisms are compartmentalized in 3D-printed microchambers surrounded by 0.22-μm filters, thereby creating closed gnotobiotic containers from which metabolites (Meta), but not organisms, can exit at varying flow rates (Flow). The metabolites can either be collected directly or can travel through subsequent microchambers connected in series to affect the growth and metabolism of other organisms via a so-called "train" system (Train). The MetaFlowTrain is cheap, reusable, multichannel, modular, and can accommodate a wide range of organisms.

One MetaFlowTrain system consists of four different compartments:

1. One or multiple bottle(s) containing sterile media that can accommodate versatile liquid inputs (Fig. 1a) such as artificial growth media (e.g., TSB, ARE), natural extracts (e.g., soil or plant extracts), as well as exudates previously obtained from a given organism. Notably, up to 24 different liquid input media can be used in parallel within each MetaFlowTrain system, thereby opening the possibility to test how media perturbations modulate organismal growth and exometabolite production.

2. A 24-channel peristaltic pump (Fig. 1b) imposes a directional flow in 24 independent/parallel metabolic trains by flushing the input liquid media through the gnotobiotic microchambers. The flow rate is fully adjustable, ranging from 7.3 μL/min to 7.3 mL/min (Supplementary Fig. 2a), thereby allowing the use of as little as 10.5 mL of medium per train and per day. However, we noticed that the real flow rate was slightly lower compared to the expected flow rate defined on the pump (21.35% reduction on average) and that this difference was consistent across various flow rates ($R^2 > 0.99$, Supplementary Fig. 2a). The number of trains that can run in parallel is defined by the number of channels of the peristaltic pump and, therefore, the throughput can easily increase to 96 independent metabolic trains with four 24-channel peristaltic pumps (i.e., four MetaFlowTrain systems).

3. A modular microchamber "train" system (Fig. 1c) that represents the key innovation for MetaFlowTrain development. Each metabolic train described above can contain from one to six 3D-printed microchambers connected in series, thereby forming a microchamber "train" through which metabolites can travel from one microchamber to the following and so forth. Because the microchambers are surrounded by 0.22-μm filters, they are independent and fully gnotobiotic, can accommodate organisms from different kingdoms of life such as bacteria or fungi (see below) as well as photosynthetic microalgae (Supplementary Fig. 2b), and are therefore suitable for studying exometabolite-mediated interactions across a broad range of conditions. We validated that "trains" with up to six different microchambers have stable flow rates, irrespective of the "train" size (Supplementary Fig. 2a) and we established a cleaning protocol, making this system reusable, cheap, and sustainable (see methods).

4. Multiple temporal outputs can be collected. Microbial cells are harvested through the hole at the top of each microchamber, while metabolite collection is performed from the 15 mL Falcon tubes at the end of each metabolic train (Fig. 1d). Outputs are multifaceted, enabling microbe collection inside the microchambers and various other analyses, such as OD measurement, quantitative PCR, and amplicon sequencing. Additionally, metabolites obtained in the collection tube can be used for targeted and untargeted metabolomics but also for various biological assays.

To facilitate user understanding of the MetaFlowTrain system, we provide a detailed movie explanation (Supplementary Movie 1), a visual summary (Supplementary Fig. 1), and a demonstration of the system in operation in the lab (Supplementary Movie 2). Additionally, a protocol (including Materials, Procedure, and Troubleshooting) is available in PDF format (Supplementary Protocol 1), with the latest version accessible on protocols.io, where future updates will also be provided (https://doi.org/10.17504/protocols.io.36wgqd68ovk5/v3). For all 3D-printed objects used in this study, we provide the complete set of 3D printing models (Supplementary Data 1[24]) and individual stereolithography files (Supplementary Data 2[24]), to enable easy setup of the MetaFlowTrain system.

In the following sections, we present proof-of-concept experiments and results obtained using MetaFlowTrain that illustrate the broad potential of the method in the context of the plant microbiota.

### MetaFlowTrain is a microbial growth system suitable for collecting microbially-derived exometabolites
The primary function of MetaFlowTrain is to cultivate a wide range of phylogenetically distinct microbes and to collect microbially-derived exometabolites. By continuously feeding microorganisms with fresh media, MetaFlowTrain largely minimizes the confounding effects of medium consumption and metabolite production observed in closed systems and, therefore, should more closely mirror natural nutritional fluxes. Here, we used phylogenetically diverse bacterial (n = 11) and fungal (n = 4) strains representing the core microbiota colonizing *Arabidopsis* roots across Europe[2]. Rather than using rich and/or artificial growth media, we prepared a sterile peat extract (see Methods for details) that better mimics complex soil nutritional inputs. Next, we inoculated each strain individually in different microchambers or left them empty (16 conditions, 3 replicates, 51 trains, 1 microchamber per train), applied a continuous nutrient flow rate of 7.3 μL/min, and collected both the microbes and their exometabolites 62 h post inoculation (hpi) (Fig. 2a) (Please refer to the Methods section for detailed information on our approach to harvesting both microbes and exometabolites).

Although the sterile peat extract largely failed to drive microbial proliferation in classical 96-well plates, it promoted microbial growth

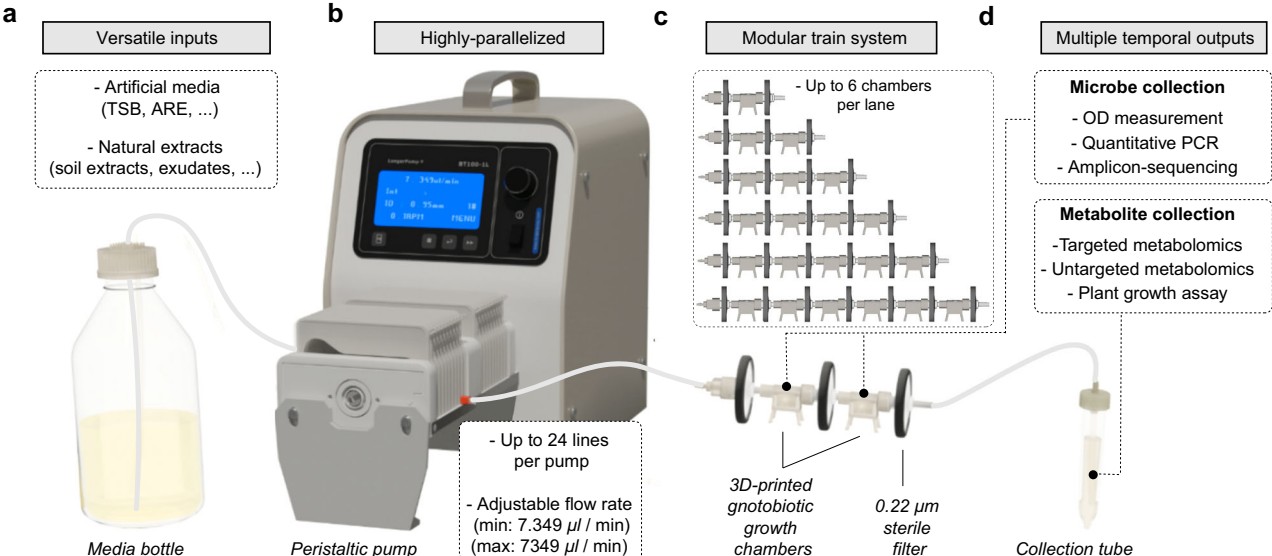

**Fig. 1 | MetaFlowTrain system overview. a** MetaFlowTrain supports diverse inputs, including artificial media (e.g., TSB, ARE) and natural extracts (e.g., soil extracts, host exudates). **b** The system operates up to 24 trains per peristaltic pump with adjustable flow rates. **c** A modular setup allows for up to six microchambers per train. **d** Outputs include microbial and exometabolite collection for OD measurements, qPCR, amplicon sequencing, metabolomics, and plant growth assays. A movie describing the system is also available (Supplementary Movie 1).

in MetaFlowTrain (Supplementary Fig. 2c), with nine bacterial and two fungal strains that actively grew based on $OD_{600}$ measurements between 0 and 62 hpi (Wilcoxon-test, $P < 0.05$, Fig. 2b). Our results suggest that continuous nutrient supply was important for driving microbial proliferation in nutrient-poor media such as peat extract. Metabolic profiling of metabolite samples collected at 62 hpi using both untargeted and targeted approaches (see Methods), revealed significant differences in metabolic profiles among strains, with 37.2% of variation explained by the different strains for untargeted profiles (PERMANOVA, $P < 0.001$, Fig. 2c) and 49.8% for targeted profiles (PERMANOVA, $P < 0.001$, Fig. 2c). Our results indicate that MetaFlowTrain supports the growth of phylogenetically distinct strains belonging to different microbial kingdoms and enables differentiation of exometabolite profiles between strains.

## MetaFlowTrain reveals soil conditioning effects on SynCom structure and allows determination of the impact of exometabolites on plant growth

MetaFlowTrain can accommodate multiple nutrient inputs and is designed for SynCom reconstitution experiments. In this study, we used the same bacterial and fungal strains defined above and assembled them into a 15-member Bacterial-Fungal SynCom (named hereafter BF SynCom). We asked whether peat soil conditioning by SynCom and/or *Arabidopsis* alters soil metabolic profiles and affects BF SynCom structure and exometabolite production in the microchambers. Sterile peat extracts obtained from peat, peat-containing *Arabidopsis*, or peat-containing *Arabidopsis* and SynCom (see Materials and Methods for details) were flushed at 7.3 μL/min into microchambers containing the BF SynCom or left sterile (6 conditions, 7 replicates, 42 trains, 1 microchamber per train). At 62 hpi, the microbes were collected for amplicon sequencing, and their associated exometabolites were analyzed using targeted metabolomics (Fig. 3a).

Sterile extracts derived from different peat conditioning treatments differentially modulate bacterial (81% of variance, PERMANOVA, $P < 0.001$) but not fungal (23.4% of variance, PERMANOVA, $P = 0.12$) community composition in the microchambers (Fig. 3b). These differences were primarily driven by the extract obtained from peat conditioned by both *Arabidopsis* and the SynCom and were associated with a significant depletion of six bacterial strains compared to control

peat extract (Kruskal–Wallis test and *post-hoc* Dunn test, $P < 0.05$) (Fig. 3c). A similar, yet not significant reduction was observed for all four fungal isolates, suggesting that *Arabidopsis* with SynCom members have left a metabolic footprint in the peat soil that extensively modulates BF SynCom assembly in the microchambers. This was corroborated by targeted metabolic profiles of samples collected at the end of the metabolic trains (24 hpi and 62 hpi), identifying the impact of the "Medium" effect (51.5%, PERMANOVA, $P < 0.001$) as greater than both "SynCom" effect (27.2%, PERMANOVA, $P < 0.001$) or "Time" effects (3.5%, PERMANOVA, $P < 0.002$) on metabolic profile differentiation (Fig. 3d). It is noteworthy that no metabolic differentiation was observed over time for mock samples, whereas a temporal shift was observed for BF SynCom-inoculated samples (Fig. 3d), thereby validating our MetaFlowTrain approach. Compared to mock-inoculated microchambers, those inoculated with the BF SynCom showed complete depletion of alpha-ketoglutaric acid and fumaric acid ($Log_2FC = -5$), indicating that these molecules, present in all peat extracts, have been consumed by the BF SynCom (Fig. 3e). This differential enrichment analysis also revealed significant differences across peat extracts (Kruskal–Wallis test and *post-hoc* Dunn test, $P < 0.05$). This was again primarily explained by the medium obtained from peat being conditioned by both *Arabidopsis* and the SynCom, which exhibited higher $Log_2FC$ for certain organic acids and sugar alcohols [2PG (2-Phosphoglyceric acid) and 3PG (3-Phosphoglyceric acid), succinic acid, malic acid, mannitol, sorbitol] and amino acids (Aspartic Acid, Proline) and lower $log_2FC$ for maltose/sucrose compared to control peat extract (Fig. 3e).

The collected exometabolites were also tested on plant growth using an agar plate-based assay (see Methods for details). Plants exposed to exometabolites obtained from SynCom-inoculated microchambers in general promoted plant germination (15% higher germination rate in average, Fisher's Exact Test, $P < 0.01$, Fig. 3f) and *Arabidopsis* root length (6.35 mm longer in average, Wilcoxon Test, $P < 0.05$, Fig. 3g) compared to those exposed to exometabolites derived from mock-inoculated microchambers. This beneficial effect of microbially-derived metabolites on plant traits was, however, dependent on the peat extract used since no significant promotion of seed germination and root growth was observed when the peat was conditioned by *Arabidopsis* and the SynCom ($P = $ ns) (Fig. 3f, g). Our results indicate that

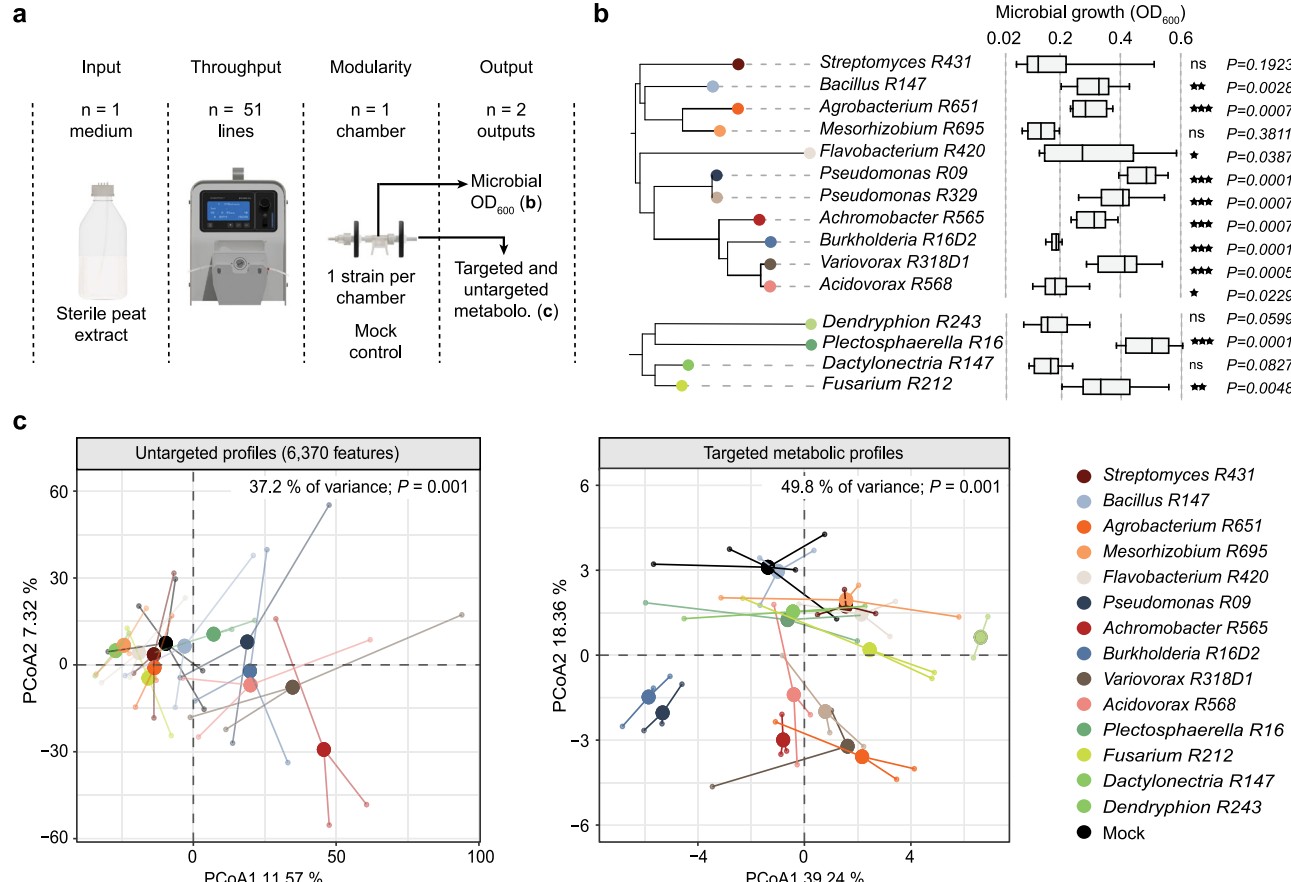

**Fig. 2 | Growth and exometabolite profiling of *Arabidopsis* root microbiota using MetaFlowTrain. a** Schematic of MetaFlowTrain setup: sterile peat extract as input, 51 parallel trains, one microchamber per train, and two outputs: microbial growth ($OD_{600}$) and exometabolite profiles (targeted/untargeted). **b** Neighbor-joining phylogenetic trees of BF SynComs members (n = 11 bacteria, n = 4 fungi). Boxplots display individual strain growth after 62 hours ($OD_{600}$). Boxplots are delimited by the first and third quartiles, with the central line representing the median value. Whiskers extend to show the range of the data within 1.5 * IQR from the quartiles. n = 3 independent microchambers per strain. Significant differences compared to inoculum OD as control was determined with two-sided Wilcoxon tests (* $P < 0.05$, ** $P < 0.01$, *** $P < 0.001$). **c** PCA plots of untargeted (left, n = 6370 features) and targeted (right, n = 35 metabolites) metabolic profiles for each microbial strain. n = 3 independent microchambers per strain, except *Variovorax*, n = 2 independent microchambers per strain, and Mock, n = 5 independent microchambers per strain. Statistical significance was determined by PERMANOVA-based comparison of metabolic profiles of individual strains (CanberraDistances-Strains, permutations = 999). Colors represent the different strains. Source data are provided as a Source Data file.

the metabolic legacy imprinted in soil following conditioning modulates SynCom assembly and plant health through exometabolite production. They also illustrate the power of MetaFlowTrain to simultaneously study media input composition, SynCom assembly processes, and the associated metabolite production that drives plant performance.

We next tested whether germ-free *Arabidopsis* could be directly connected to MetaFlowTrain and whether root-exuded metabolites can modulate microbial growth in the microchambers. We pumped ½ MS medium in a peat-based semi-hydroponic system that either contained 6-week-old *Arabidopsis* or were left unplanted. We funneled these conditioned media directly into microchambers containing the BF SynCom over the course of 6 days (Supplementary Fig. 3a). At the end of the experiment, we observed that exudate-conditioned media differed across conditions (66.1%, PERMANOVA, $P = 0.002$) (Supplementary Fig. 3b) and this translated into differential effects on BF SynCom growth in the microchamber based on qPCR measurements (Supplementary Fig. 3c). MetaFlowTrain may therefore also allow the study of host-derived exometabolites and how they shape microbiota assembly.

### MetaFlowTrain reveals strong inhibitory activity of bacterial exometabolites towards fungi
We next tested the relevance of the multi-chamber train system for unraveling exometabolite-mediated microbial interactions. Using our

BF SynCom, we tested whether microbes that evolved in different kingdoms of life impact each other's growth through exometabolite production. We used sterile peat extract as an input and flushed this medium at a flow rate of 7.3 μL/min into microchamber trains consisting of two microchambers connected in series. We used either F SynCom−B SynCom, B SynCom−F SynCom, or Mock−BF SynCom train combinations (3 conditions, 7 replicates, 28 trains, 2 microchambers per train) (Fig. 4a). Therefore, the exact same strains were used in the different trains but the order or physical contact differed between B and F SynComs, thereby facilitating an understanding of the directional impact of bacterial and fungal exometabolites on F and B SynComs, respectively. At 62 hpi, the microbes were collected for amplicon sequencing, and their associated exometabolites were analyzed using targeted metabolomics (Fig. 4a).

We did not observe a significant impact of F SynCom on B SynCom assembly in any of the two configurations, while B SynCom significantly impacted Fungal SynCom abundance (Kruskal−Wallis test and *post-hoc* Dunn test, $P < 0.05$) (Fig. 4b). We observed a very strong, significant, and consistent decrease in the abundance of F SynCom members upon co-inoculation with B SynCom members (Mock − BF SynCom). *Fusarium R212* growth was on average reduced by 79.7%, that of *Plectosphaerella R16* by 47.5%, *Dactylonectria R147* by 59.5%, and that of Dendryphion *R243* by 82.1% (Kruskal−Wallis test and *post-hoc*

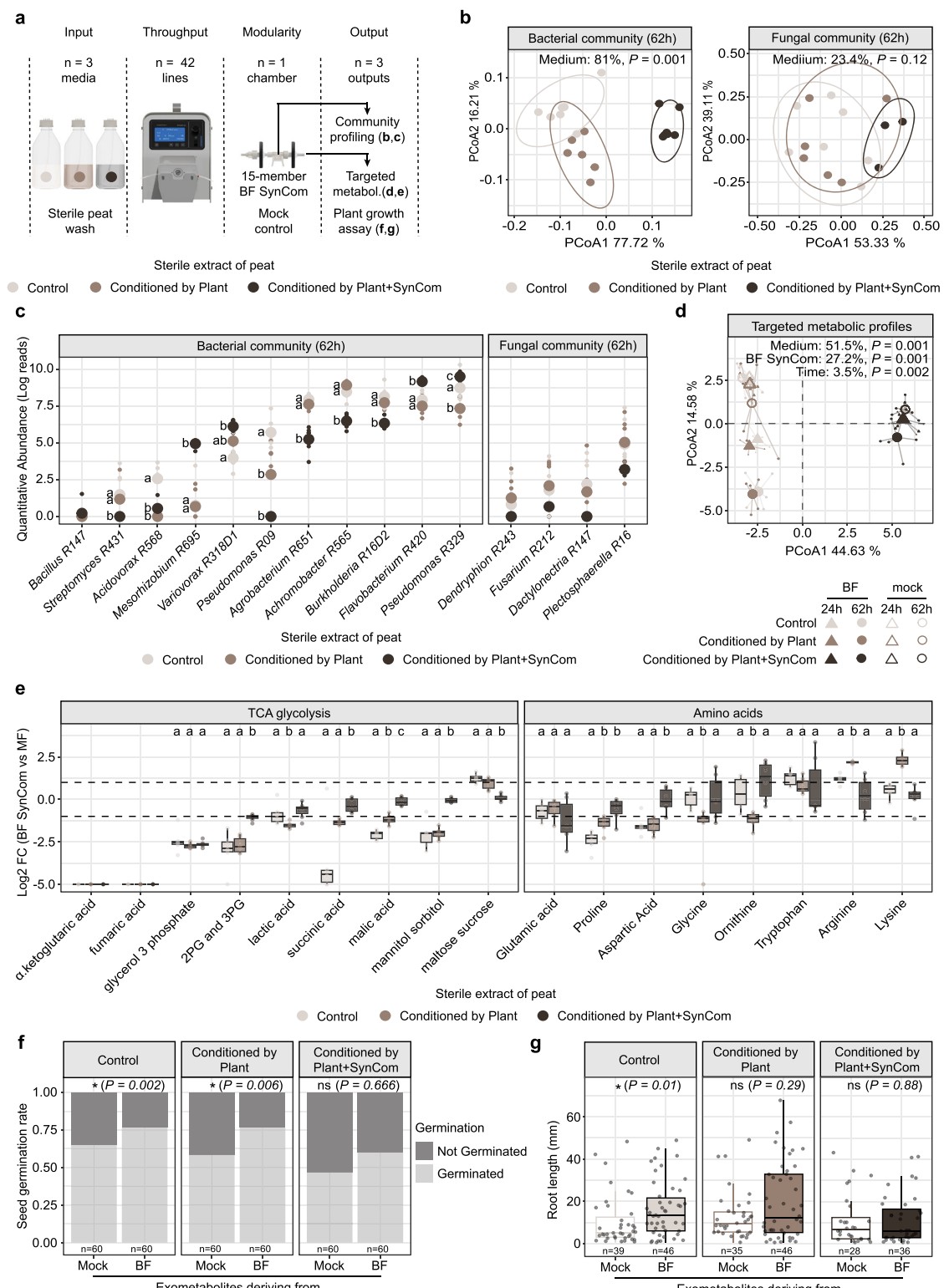

Dunn test, $P < 0.05$) (Fig. 4c). Notably, similar inhibition values (i.e., 68.4%, 36.7%, 52.7%, 67.2%, respectively) were observed when fungal isolates were exposed to B SynCom-derived exometabolites (B Syn-Com – F SynCom) (Kruskal–Wallis test and *post-hoc* Dunn test, $P < 0.05$) (Fig. 4c). The data indicate that physical contact between bacteria and fungi is largely dispensable for bacteria-mediated fungal inhibition and primarily occurs through production of inhibitory

exometabolites. The similarity in metabolic profiles across output samples, regardless of the strain combinations inoculated into the microchambers (PERMANOVA, 6.5% variation, $P = 0.46$) (Fig. 4d), support the conclusion that differential media consumption was not a driver of the fungal load reduction. However, the significant changes in metabolic profiles over time (PERMANOVA, 34.5% variation, $P < 0.001$) (Fig. 4d) confirm that microbes were actively growing within the

**Fig. 3 | Effects of soil conditioning on SynCom assembly, metabolic profiles, and plant phenotypes. a** Schematic of MetaFlowTrain setup with three different sterile extracts as inputs. 42 parallel trains, one microchamber per train, and three outputs: community profiles (MetaBarcoding), exometabolic profiles (targeted), and plant phenotyping. **b** PCA based on Bray−Curtis distances showing bacterial (left) and fungal (right) beta diversity. n = 6 independent microchambers, except for Control n = 7 independent microchambers. Statistical significance was determined by PERMANOVA-based comparison of community profiles (BrayDistances-Medium, permutations = 999). **c** Dot plot showing quantitative abundance (log reads) of each SynCom member. n = 6 independent microchambers per treatment, except for Control n = 7 independent microchambers per treatment. Letters indicate statistically significant differences between extracts (Kruskal−Wallis test ($P < 0.05$), followed by two-sided Dunn's post hoc tests with Benjamini-Hochberg correction for multiple comparisons ($P < 0.05$). **d** PCA plots of targeted exometabolite profiles (n = 35 metabolites). n = 6 independent microchambers per treatment, except for Control n = 7 independent microchambers. Statistical significance was determined by PERMANOVA-based comparison of metabolic profiles (CanberraDistances-Medium/Time/BFSynCom, permutations = 999). **e** Boxplots showing Log$_2$ fold change (Log2FC) of exometabolites in the BF SynCom compared to a control. Boxplots are delimited by the first and third quartiles, with the central line representing the median value. Whiskers extend to show the range of the data within 1.5 * IQR from the quartiles, and all data points are displayed as individual points, including outliers. Only significant exometabolites are shown. n = 6 independent microchambers per treatment, except for Control n = 7 independent microchambers. Letters indicate statistically significant differences between extracts (Kruskal−Wallis test ($P < 0.05$), followed by two-sided Dunn's post hoc tests with Benjamini-Hochberg correction for multiple comparisons ($P < 0.05$). **f** Bar chart showing germination rates (% germinated vs. non-germinated seeds) in exometabolite-derived media. n = 60 individual plants per treatment. Stars denote significance (two-sided Fisher's Exact test (* $P < 0.05$, ns = non-significant). **g** Boxplots showing root lengths (mm) of germinated plants in exometabolite-derived media. Boxplots are delimited by the first and third quartiles, with the central line representing the median value. Whiskers extend to show the range of the data within 1.5 * IQR from the quartiles, and all data points are displayed as individual points, including outliers. n = number of independent germinated plants. Significance, compared to Mock, was determined with two-sided Wilcoxon tests (* $P < 0.05$, ns = non-significant). Colors represent the different peat extracts used as media. Source data are provided as a Source Data file.

system. Our results are consistent with previous work showing the critical role of bacteria in preventing fungal dysbiosis in *Arabidopsis* roots[11,25,26] and illustrate the potential of MetaFlowTrain for uncovering microbe−microbe interaction phenotypes mediated through exometabolite production.

## MetaFlowTrain validates the inhibitory impact of key bacterial exometabolites on SynComs

To further test whether microbially-derived exometabolites, and not bacteria-mediated medium consumption, explains microbe−microbe interaction phenotypes in MetaFlowTrain, we used the root microbiota member *Pseudomonas brassicacearum* R401 wild-type (WT) and a triple mutant strain (*phld/pvdy/brpc*), in which the production of three exometabolites with inhibitory activity is impaired[13]. *PhlD*, *PvdY*, and *BrpC* are involved in the production of the antimicrobial diacetylphloroglucinol (DAPG), the iron-chelating molecule pyoverdine, and the pore-forming cyclic lipopeptide brassicapeptin, respectively. These molecules co-function in antagonizing a broad range of root microbiota members and in promoting R401 competitiveness on roots[13,26,27]. To demonstrate that the antagonistic effect of this strain is solely due to inhibitory metabolites rather than other confounding factors, we inoculated either the mutant, the wild-type (WT), or a control (no inoculation) into the first microchamber, followed by the B or F SynCom in the second microchamber (Fig. 5a). We hypothesized that the inhibitory exometabolites produced by R401 WT in the first microchamber would hinder SynCom growth in the second microchamber, and that this growth inhibition would be alleviated when the *phld/pvdy/brpc* triple mutant was used.

Inoculation of R401 WT in Chamber 1 significantly inhibited growth of both B and F SynComs in Chamber 2. This inhibition is illustrated by lower median *Ct* values of −2.68 for the B SynCom and −4.34 for the F SynCom compared to the controls, which involved Mock inoculation in Chamber 1 and B or F SynComs in Chamber 2 (Kruskal−Wallis test followed by Dunn's *post-hoc* test, $P < 0.05$). Additionally, this effect was shown to be entirely dependent on *PhlD*, *PvdY*, and *BrpC* (Fig. 5b). Notably, R401 WT and the *phld/pvdy/brpc* mutant exhibited similar growth in Chamber 1, indicating that these mutations did not affect the strain's in vitro growth. Furthermore, SynCom growth observed in Chamber 2 was comparable, regardless of whether Chamber 1 was left sterile or inoculated with the *phld/pvdy/brpc* mutant (Fig. 5b). This largely excludes the possibility that medium consumption mediated by the *phld/pvdy/brpc* mutant in Chamber 1 contributes to the inhibition of SynCom growth in Chamber 2. Our results therefore confirm that these three inhibitory exometabolites produced by R401 are causally linked to the inhibitory phenotypes.

These results also highlight MetaFlowTrain as an ideal system to disentangle confounding effects, test causality, and uncover mechanisms behind SynCom assembly through exometabolite production and exchange.

## Discussion

In this paper, we introduce MetaFlowTrain (Fig. 1 and Supplementary Movie 1), a versatile and cost-effective gnotobiotic fluidic system designed for the dynamic study of exometabolite-mediated interactions between organisms that play pivotal roles in shaping ecosystem assembly, diversity, and functioning[28–31].

We demonstrated that MetaFlowTrain efficiently accommodates a broad range of microorganisms, including bacteria, fungi, and microalgae, as well as more complex communities (Fig. 2, Supplementary Fig. 2b, c). This enables researchers to work at various ecological scales, ranging from single strains to complex multi-kingdom communities, and provides an important tool not only for collecting metabolites over time but also for testing how a broad range of organisms interact with, and respond to, chemical signals. This is particularly relevant given the increasing emphasis in chemical ecology on deciphering complex, multi-species interactions[32]. Such chemically-mediated higher-order interactions are crucial for understanding coexistence in microbiomes but also for explaining host−microbe relationships and co-dependencies[33–36].

We also present proof-of-concept data illustrating the power of MetaFlowTrain for understanding how various inputs and growth media can affect microbial community assembly but also exometabolite production of microbes (Fig. 3, Supplementary Fig. 3). In particular, we provide evidence that subtle metabolic changes in soil following conditioning by plants and microbes are sufficient to shape microbiota assembly in the gnotobiotic microchambers, to modulate exometabolite profiles released outside the microchambers, and that this has remarkable consequences for plant phenotypes (Fig. 3). Our results corroborate that microbially-derived metabolites potentially represent an untapped resource of bioactive molecules with modulatory functions[37,38], with important implications for medicine and agriculture[39–41]. MetaFlowTrain can, therefore, help to unravel the complex metabolic interplay linking input chemical composition with production of microbially-derived exometabolites and promote discovery of new bioactive molecules.

We further provide evidence that MetaFlowTrain is suitable for deconstructing complex ecosystems by compartmentalizing organisms and uncoupling chemical and physical interactions. This is achieved via our innovative microchamber train system that allows for the biological deconstruction of complex communities into sub-

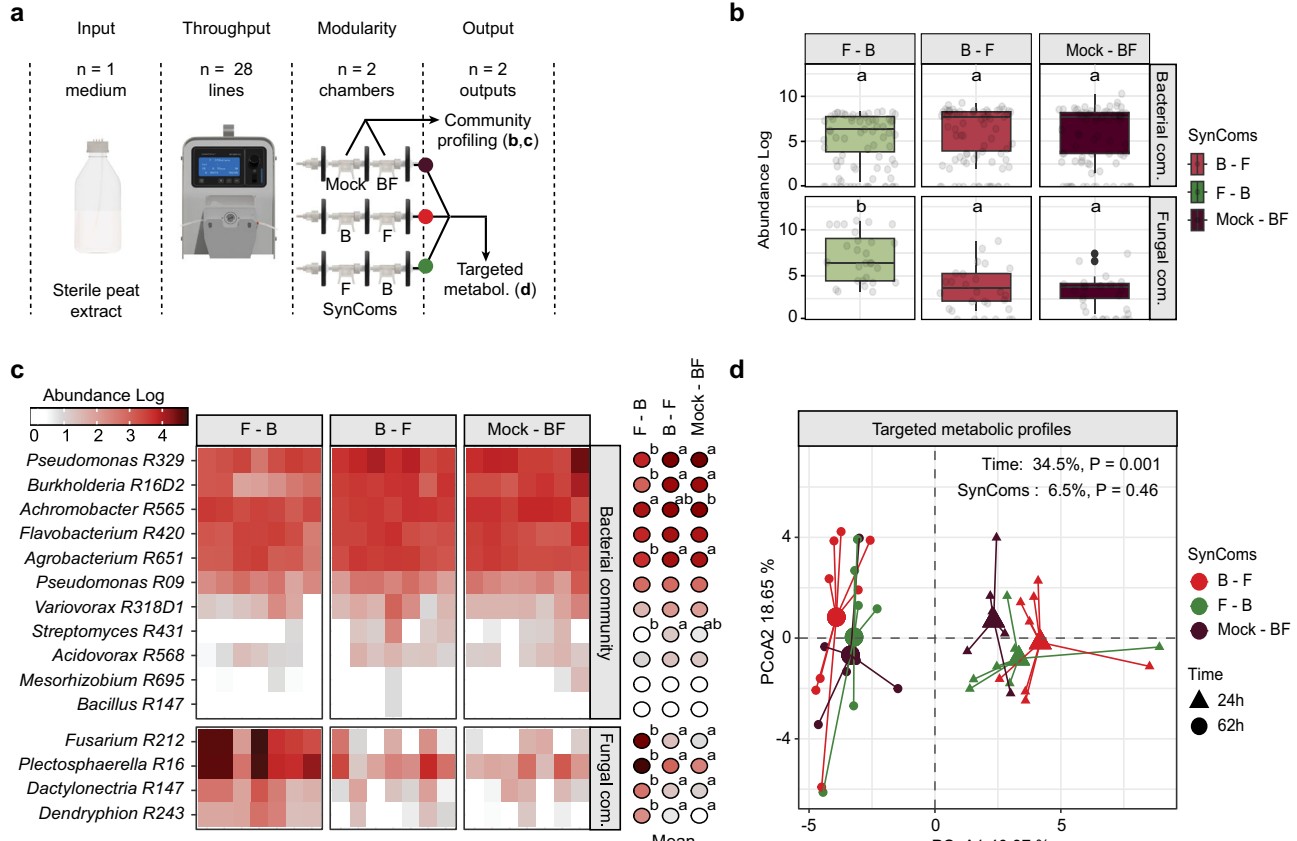

**Fig. 4 | Impact of bacterial and fungal exometabolites on SynCom structure and metabolic profiles. a** Schematic of the MetaFlowTrain setup with sterile extract as input, 28 parallel trains, two microchambers per train, and two outputs: community profiles (MetaBarcoding) and exometabolites (targeted metabolomics). **b** Boxplots showing the abundances of microbial community members (log₁₀ reads) for different microchamber combinations. Boxplots are delimited by the first and third quartiles, with the central line representing the median value. Whiskers extend to show the range of the data within 1.5 * IQR from the quartiles, and all data points are displayed as individual points, including outliers. n = 7 independent microchamber trains * n = 11 bacteria (n = 77) or n = 4 fungi (n = 28) per treatment. Letters denote statistically significant differences between bacterial abundance or fungal abundance (Kruskal–Wallis test ($P < 0.05$), followed by two-sided Dunn's post hoc tests with Benjamini-Hochberg correction for multiple comparisons ($P < 0.05$)). **c** Heatmap showing bacterial and fungal community composition across microchamber combinations. Red color intensity indicates

log-transformed absolute abundances. Balloon plot shows the mean absolute abundance of microbial community members. n = 7 independent microchamber trains per treatment. Letters indicate statistically significant differences (Kruskal–Wallis test ($P < 0.05$), followed by two-sided Dunn's post hoc tests with Benjamini-Hochberg correction for multiple comparisons ($P < 0.05$). **d** PCA plots displaying targeted metabolic profiles (n = 35 metabolites). n = 7 independent microchamber trains per treatment, except for B–F, n = 8 independent microchamber trains. Statistical significance was determined by PERMANOVA-based comparison of metabolic profiles (CanberraDistances-Time/SynComs, permutations = 999). The microchamber configuration is denoted as Microchamber1–Microchamber2 (e.g., B–F: Bacterial SynCom in microchamber 1 and Fungal SynCom in microchamber 2). Colors represent the different combinations of SynCom used in the microchambers. Source data are provided as a Source Data file.

communities that are metabolically but not physically connected. This compartmentalization capability, therefore, resolves a common limitation in closed systems where metabolites are mixed and cannot be attributed to specific species[42]. Consistent with previous work that used similar root microbiota members[11,13,25], we observed that bacteria exert strong inhibitory effects towards fungi and that fungi have comparatively less influence on bacterial assembly (Fig. 4). In Meta-FlowTrain, we revealed that these effects do not require physical contact and occur through exometabolite production. The data illustrate that MetaFlowTrain has high potential to screen and identify microbes and associated molecules that have novel, potent, or highly specific antimicrobial activities towards important plant and animal pathogens.

By controlling the direction of metabolic fluxes within the microchamber(s), the system enables the detection of directional effects. Furthermore, while medium consumption still occurs within the microchambers, the continuous influx driven by the peristaltic pump significantly reduces the confounding effects of medium

consumption versus metabolite production, which are common in closed systems. Indeed, we provide evidence that exometabolite production, and not medium consumption is causally linked to the inhibitory phenotypes that we observed. This is illustrated by our observation that a bacterial triple mutant impaired in the production of three inhibitory exometabolites[13,26] grew extensively in Chamber 1 and that this was not associated with a significant reduction of microbial growth in Chamber 2 (Fig. 5). Therefore, MetaFlowTrain represents a powerful growth platform to uncouple metabolite consumption and production and to study causality in microbe–microbe interactions.

MetaFlowTrain offers exciting opportunities, yet the method does have some limitations. It is a liquid-based system that requires previous collection of soil or host extracts and that overlooks spatial and physical heterogeneity often observed in natural ecosystems[43,44]. The unidirectional flow can be viewed as an advantage of the system to understand causality but represents at the same time a weakness because it overlooks bi-directional exchange, inducible mechanisms,

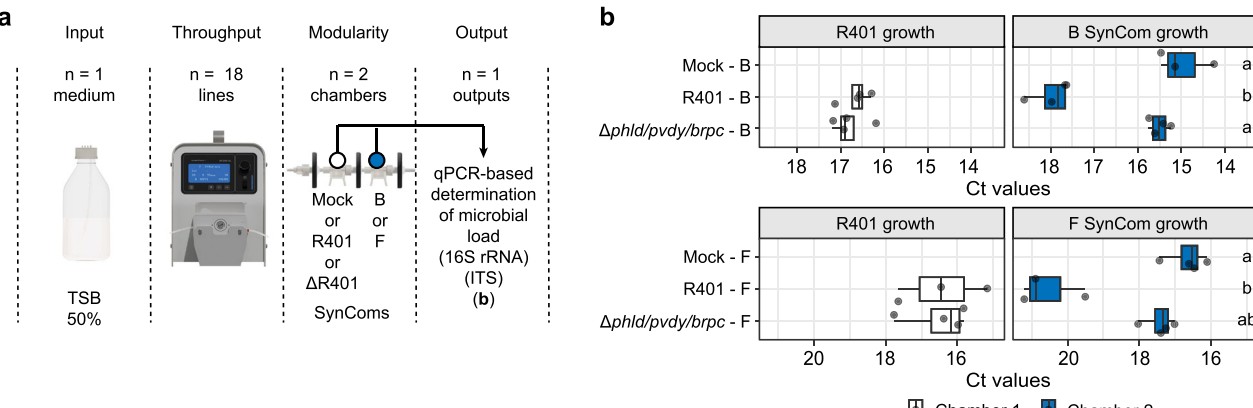

**Fig. 5 | Impact of *Pseudomonas brassicacearum* exometabolites on SynCom growth. a** Schematic of the MetaFlowTrain setup: TSB 50% as input, 18 parallel trains, two microchambers per train, with one output: microbial load (qPCR). **b** Boxplots showing microbial load (*Ct* values) for different combinations of SynComs (Bacterial, Fungal, Mock) and R401 WT or mutant. Boxplots are delimited by the first and third quartiles, with the central line representing the median value. Whiskers extend to show the range of the data within 1.5 * IQR from the quartiles, and all data points are displayed as individual points, including outliers. n = 4 independent microchamber trains per treatment, except for R401–F and R401–B, n = 3 independent microchamber trains. Letters denote significant differences between microchambers (Kruskal–Wallis test (*P* < 0.05), followed by two-sided Dunn's post hoc tests with Benjamini-Hochberg correction for multiple comparisons (*P* < 0.05)). The microchamber configuration is denoted as Microchamber1–Microchamber2 (e.g., Mock–B: Mock in microchamber 1 and Bacterial SynCom in microchamber 2). Colors represent the different Chambers based on their position. Source data are provided as a Source Data file.

and more complex feedback loops that sometimes occur between cells or organisms[45,46]. Although it is possible to create circular lines (i.e., closed), we have not yet explored this possible modular aspect of the system. The system was not initially developed to be connected to a living host. However, we provide here proof-of-concept data illustrating that exudates of sterile plants can be collected by MetaFlowTrain and directly flushed into microchambers to modulate SynCom growth (Supplementary Fig. 3). Therefore, the system also has high potential for reassembling complex plant–microbiota synthetic ecosystems that run autonomously through metabolic interactions. Although not tested in this study, the MetaFlowTrain system is expected to function under anaerobic conditions, as its compact design is compatible with standard anaerobic microchambers.

We envision that MetaFlowTrain will have broad applications in both academic and industrial research, serving as a powerful tool to address diverse questions across fields such as environmental ecology, microbiology, biotechnology, plant biology, or chemistry. By applying various controlled perturbations in the input compartment (i.e., nutrients, exudates, chemicals), in the set-up of the peristaltic pump (flux rate), within or between microchambers (type and diversity of microorganisms, including genetically modified organisms), the system is highly suitable for mechanistic studies as well as for those aiming at establishing causalities.

## Methods
### Microbial strains
The 15 strains from the SynCom, R401 WT, and *Chlamydomonas reinhardtii CC1690* used for this study were previously reported[47–49]. The R401 triple mutant is available from the Max Planck Institute for Plant Breeding Research upon request from Stéphane Hacquard (hacquard@mpipz.mpg.de). All strains are summarized in Supplementary Table 1.

### Microbial inoculum preparation
Bacterial strains were cultivated from their glycerol stocks (stored at −80 °C in 50% glycerol) on 50% Tryptic Soy Agar (TSA, Sigma-Aldrich) for six days at 25 °C and then grown in Tryptic Soy Broth (TSB, Sigma-Aldrich) overnight before inoculation. The bacterial cultures were centrifuged at 2000 × *g* for 10 min and resuspended in 10 mM

Magnesium chloride (MgCl₂, Sigma-Aldrich) to remove residual media and bacterial metabolites. The optical density (OD) of each bacterial strain was adjusted according to the experiment. For SynCom, strains were pooled in equal ratios.

Fungal strains, stored as pieces of mycelium in 30% glycerol at −80 °C, were individually cultivated on Potato Glucose Agar (PGA, Sigma-Aldrich) for two weeks. The mycelium was harvested using sterile tips into 1 mL of MgCl₂ containing a stainless-steel bead with D = 2.85–3.45 mm (Carl Roth) and 10 stainless steel beads with D = 0.75–1 mm (Carl Roth) and crushed with 1 mL 10 mM MgCl₂ in a paint shaker (SK450, Fast & Fluid Management, Sassenheim, Netherlands) for 10 minutes. The weight of each fungal strain was adjusted according to the experiment. For SynCom strains were pooled in equal ratios. *Chlamydomonas reinhardtii CC1690 (Cr)* cells were grown photoautotrophically in TP10[50] at 25 °C, and the illumination of 125 µmol m⁻² s⁻¹ under continuous light conditions. Cultures were kept in a rotatory at 70 RPM. On the day of the experiment, cell concentration was determined by measuring samples in a Multisizer 4e Coulter (Beckman Coulter Inc., California, USA) particle counter with the Beckman Coulter Multisizer software (v4.03). *Cr* concentration inoculum was adjusted using TP10 media at 2.5 million cells per mL.

### Plant growth conditions
*Arabidopsis thaliana* Col-0 wild-type (N60000) was obtained from the Nottingham *Arabidopsis* Stock Centre (NASC). *Arabidopsis thaliana* Col-0 seeds were sterilized using 70% ethanol and bleach. The seeds were submerged in 70% ethanol and shaken at 40 rpm for 15 min (rotator SB3, Stuart). After removing the ethanol, the seeds were submerged in 8.3% sodium hypochlorite (Carl Roth) containing 1 µl of Tween 20 (Sigma-Aldrich) and shaken at 40 rpm for 5 minutes. Under sterile conditions, the seeds were washed seven times and finally suspended in sterile 10 mM MgCl₂. The seeds were left for stratification at 4 °C for 3 days. Seed sterility was confirmed by plating approximately 100 seeds on 50% TSA plate.

### Sterile peat extracts using gnotobiotic FlowPot system
Different sterile peat extracts were prepared using the FlowPot system[8]. For the peat extracts composed of plant exudates and microbial exudates, bacterial strains were cultivated from their

glycerol stocks (stored at −80 °C in 50% glycerol) on 50% Tryptic Soy Agar, for six days at 25 °C, and then grown in Tryptic Soy Broth, overnight before inoculation. Each individual bacterial inoculum and fungal inoculum was adjusted to an OD of 0.5 and a weight of 50 mg/mL. Thereafter, strains were pooled in equal ratios to form the Bacterial SynCom and the Fungal SynCom. The gnotobiotic FlowPot system was prepared by adding glass beads to the Luer end of a truncated 50 mL syringe, followed by the addition of twice-autoclaved peat. The peat was then covered with a mesh retainer and secured with a cable tie. FlowPots were subsequently autoclaved before microbial inoculation[8]. Microbial mixtures were adjusted to a biomass ratio of 4:1 (eukaryotes:prokaryotes[51]), using 200 μL (OD$_{600}$ = 0.5) of bacterial inoculum and 200 μL of fungal inoculum (50 mg/mL) in 50 mL of ½ MS (Murashige & Skoog Medium with Vitamins, Duchefa), and then inoculated into the FlowPot using a 50 mL syringe. Eight inoculated FlowPots were placed into each sterile microbox (TP1200, Sac O2) and stored at room temperature for two days. Exactly five sterilized seeds were sown per FlowPot and left to grow at 21 °C, with a light cycle of 10 h of light at 19 °C and 14 hours in the dark, for five weeks. Microboxes were randomized every week. For sterile peat extracts, both with and without plant exudates, the Flowpots were flushed with ½ MS solution without microbial inoculum. For sterile peat extracts without plant exudates, no seeds were sown in the Flowpots. After 5 weeks, each Flowpot was flushed with 20 mL of sterile water under sterile conditions. We selected 20 mL as it corresponds to the volume of the Flowpot. One milliliter of the flush was plated on a 50% TSA plate to check for sterility, while the remaining flush from all Flowpots with the same conditions was combined and filtered through a 0.22 μm filter unit to remove small particles.

### 3D design of the microchamber trains, connectors, and racks

Microchambers, connectors, Duran® bottle adapters, 15-mL Falcon adapters, and racks were all designed using Fusion 2.0.19440 (Autodesk).

- The MetaFlowTrain microchamber is a rectangular microchamber with dimensions of 10 mm (width), 5 mm (length), and 8 mm (height), giving it a volume of 400 μL (Supplementary Fig. 4a). The bottom surface is composed of triangular teeth 1.5 mm high with a rectangular angle on the input side. These teeth demonstrated better retention of microbes and helped prevent clogging of the filters compared to microchambers without teeth (Supplementary Fig. 4b, c). The microchamber has one female Luer fitting as an inlet and one male Luer lock fitting as an outlet, compatible with standard syringe filters. A top hole with a 2-mm luer lock fitting allows for the inoculation and harvesting of microbes. Each microchamber can be easily connected on both sides with 0.22 μm syringe filters (PVDF membrane), allowing complete compartmentalization of the microchambers.
- The MetaFlowTrain connector is a 2 cm-long connector with one female fitting (1 mm inner diameter) as an inlet and one male Luer Lock fitting as an outlet.
- The Duran® bottle adapter has a 1 mm inner diameter male outlet on both the outside and inside. In the manuscript, we describe a model with 12 male outlets.
- The 15-mL Falcon Cap adapter is a 15 mL falcon cap with a 2 mm inner diameter male inlet on top and outlet inside.
- MetaFlowTrain racks are racks designed to help handle the microchambers. The racks are composed of five supports for five microchamber trains. They can be stacked and are provided for trains with one unique microchamber (approximately 6 cm long, and for two or more microchambers (approximately 12 cm long).

All printing models (Supplementary Data 1[24]) and individual stereolithography files (Supplementary Data 2[24]) can be uploaded on our online protocol on protocol.io (https://doi.org/10.17504/protocols.io. 36wgqd68ovk5/v3) or from Zenodo (https://doi.org/10.5281/zenodo. 15020528)[24]. A file with a ".form" format can be directly uploaded to a printer interface (e.g., Preform, Formlabs) and be printed.

### MetaFlowTrain experiment

We strongly recommend referring to our online protocol for performing any MetaFlowTrain experiment. (Supplementary Protocol 1, protocol.io: https://doi.org/10.17504/protocols.io.36wgqd68ovk5/v3).

Before starting the experiments, microchambers and connectors were 3D-printed with biocompatible resin (Biomed Resin, Formlabs). Additionally, Duran® bottle adapters with 12 male inputs and outputs, as well as 15 mL Falcon adapters, were 3D-printed using Biomed Amber Resin (Formlabs). Prior to assembly of the MetaFlowTrain, all 3D models were autoclaved. We performed all prints on a Form3B+ (Formlabs), washed them with 99.9% isopropanol (Robbyrob) a first-generation Form Wash (Formlabs), and cured them with a Form Cure (Formlabs), following the manufacturer's recommendations.

The assembly of a train of microchambers was performed under sterile conditions. The assembly consisted of one MetaFlowTrain connector (Male 1 mm inner diameter, Female Luer lock) prefilled with one silicone ring (Silicone O-Ring O-Ring, 3.68 mm Bore, 9/32in Outer Diameter, RS PRO, 527-9756), screwed to one sterile 0.22 μm syringe filter (0.22 μm Syringe Filter, PVDF (Sterile), Blue, diam. 33 mm, Pk/100, Starlab, E4780-1221). The syringe filter was plugged into the male part of the microchamber, and the female part was filled with one silicone ring and a second sterile 0.22 μm syringe filter was screwed onto it. These steps were repeated according to the number of desired microchambers. Each train of microchambers for the MetaFlowTrain could be assembled before the experiment and stored in the dark, dry conditions in a sterile box (Supplementary Movie 3, 00:21:12).

On the day of the experiment, the microchambers with a volume of about 400 μL were inoculated with 40 μL of MgCl$_2$ for controls (Mock). The bacterial and fungal inocula were adjusted to an OD$_{600}$ of 0.2 and a concentration of 2 mg/mL, respectively. For single strain experiments, 40 μL of individual strains were inoculated in the microchambers. For SynCom experiments, strains were pooled in equal ratios to form the Bacterial SynCom (B) and the Fungal SynCom (F). For these single-kingdom SynComs, 20 μL were inoculated in the microchamber. For Multikingdom SynCom (BF), 20 μL of the Bacterial SynCom and 20 μL of the Fungal SynCom were inoculated in the same microchamber. *Cr* concentration was adjusted with TP10 to 2.5 million cells per mL using a Multisizer 4e Coulter (Beckman Coulter Inc., California, USA) particle counter with the Beckman Coulter Multisizer software (v4.03). We inoculated 40 μL of *Cr* in the microchambers. Microchambers were inoculated from the top hole, which was then sealed with 1 cm-wide, 2 cm-long strips of biocompatible tape (Polyolefin Diagnostic Tape, Nr 9793 R, 3 M). This hydrophobic tape can be in contact with the microbial suspension in the microchambers and has a low level of extractables in various solvents, ensuring it does not impact further metabolomics analysis; for details, refer to the manufacturer's information (Supplementary Movie 3, 03:00:00).

For all experiments, we used peristaltic pumps with three rollers and 24 channels per pump (LG-BT100-1L-A-EU/DG-24-A, Darwin Microfluidics). The peristaltic pump hoses (3-stop Platinum-cured Silicone Tubing: SE-TUB-SIL-SSS-1*1, Darwin microfluidics) were washed with 70% ethanol for 15 min, followed by sterile water for 15 min at a speed of 1000 μL per minute. A Duran® bottle adapter was screwed to the bottle of media. The inlet hoses (3-stop Platinum-cured Silicone Tubing: SE-TUB-SIL-SSS-1*1 (Darwin Microfluidics)) were connected to the Duran® bottle adapter, and the outlet hoses were connected to the connectors of the microbial trains. Outputs of the microbial trains were connected to a Tygon tube (Tygon E-LFL Pump Tubing (7.5 m): SA-AVX42007, Darwin Microfluidics) and then to a 15 mL Falcon tube using a 3D-printed adapter. Once everything was connected, the entire system was moved to a dark room at 25 °C.

Finally, the flow rate was set to 7.3 μL/min to start the experiment (Supplementary Movie 3, 03:50:00). We utilize a flow rate of 7.3 μL/min, since this is the lowest possible flow rate compatible with our pump and hose dimensions. The aim is to maximize contact time and, consequently, enhance the impact of metabolites on microbial interactions between the microchambers.

## MetaFlowTrain harvesting

For exometabolite harvesting, we empty the 15 mL Falcon tube and allow the pump to release metabolites for 4 h to reach at least 2 ml. We then transfer 2 mL into an Eppendorf tube and evaporated at 8 °C under vacuum until dry. In our study, a 2 mL concentrated sample was adequate for targeting our metabolites from TCA cycle, glycolysis, and amino acid metabolism. However, depending on the study's objectives, this volume may need to be adjusted, as it can vary based on factors such as the initial abundance of target metabolites, sample complexity, media composition, and instrument sensitivity, among others.

We performed exometabolite harvesting at 24 h and 62 h. We chose a 24 h sampling time to capture an early snapshot of the metabolomic profile, while 62 h was selected as it allowed for observing both bacterial and fungal growth. Although the experiment could have been extended, the use of peat extract, a natural rather than artificial medium, limited the volume we could produce.

For microbial harvesting, at the end of each MetaFlowTrain experiment, the pump is turned off and the entire system is moved to a sterile bench. Microchamber trains are disconnected from the pump and the Tygon tubing. The tops of the microchambers are wiped with ethanol and pierced with a sterile tweezer. Using the top hole, we thoroughly pipette up and down to create a homogeneous suspension and transfer 200 μL of the suspension to a 96-well plate. The plate is then frozen at −20 °C pending further processing.

## Washing 3D-printed parts

All 3D-printed objects can be washed and reused (e.g., microchambers, connectors, Duran® bottle adapters, and 15 mL Falcon adapters). At the end of an experiment, the microchambers, connectors, and adapters should be collected in separate glass containers. For microchambers, the tape from the top hole should be removed. The glass containers should be filled with water until the objects are fully submerged. The containers are then placed in a sonicator (Ultrasonic Cleaner, USC-T, VWR) for 15 min. The objects should be thoroughly washed using a wash bottle filled with water; for microchambers, the inside should be thoroughly cleaned from the top hole. All objects should be allowed to dry completely and stored in a box, in the dark, at room temperature until the next use (Supplementary Protocol 1). All the experiments mentioned in this paper were performed with newly printed microchambers and connectors; only the Duran bottle adapters and 15 mL Falcon adapters were reused.

## Costs for setting up a MetaFlowTrain experiment

In this estimation we did not include the cost of the 3D printer and the peristaltic pump, as these are one-time purchases that you may already have or can use for other purposes. If you don't own them, they will be the most expensive items to acquire.

We provide a table with the current prices for each individual 3D-printed part and the consumables needed to set up a MetaFlowTrain experiment with 24 units, either with 1 or 2 microchambers per train (Supplementary Table 2). Some parts, such as the Duran® bottle adapter, Falcon 15 mL adapter, and racks for microchambers and connectors, can be printed once and reused indefinitely until they break. However, because microchambers come into contact with microbes, they may need to be reprinted if damaged or if residues remain inside. To give an estimate, the first experiment will cost around €8.28 per sample. For subsequent experiments, if you wash the microchambers, the cost will drop to around €1 per sample (for filters and silicone rings). If you need to reprint the microchambers, the cost will be approximately €2 per sample. These estimates reflect current prices and may fluctuate over time; in addition, prices are subject to supplier and country-specific variations. While these estimations provide a general sense of the costs for setting up a MetaFlowTrain experiment, they should be considered as approximations.

## Microbial load measurement

Four stainless steel beads (D = 0.75–1 mm, Carl Roth) were added to each well of frozen liquid microbial suspensions stored in 96-well plates, and sealed with a non-sticky film (HJ-Bioanalytik, Art.Nr. 900320). When suspensions were thawed, samples were crushed in a bead beater (TissueLyser II, Qiagen) for 10 minutes at 1800 rpm. DNA was extracted from 15 μl of the crushed suspension with 15 μl of Buffer I (pH 12) containing 25 mM NaOH, 0.2 mM EDTA at 95 °C for 30 min; afterwards, the pH was readjusted using 30 μl of buffer II (pH 7.5) containing 40 mM Tris-HCl. Microbial loads of these samples were then measured by quantitative PCR. For each sample, we either quantified fungal load, conducted with primers ITS1F and ITS2 which target the fungal ITS1 sequence, or/and bacterial load with primers 16S-799F and 16S-1192R that target the 16SrRNA V5–V7 region (Supplementary Data 3). Each reaction was performed by mixing 5 μl of iQ™ SYBR® Green Supermix (Bio-Rad) with 0.3 μl of 10 μM forward primer, 0.3 μl of 10 μM reverse primer, 3.4 μl of water, and 1 μL of DNA template. A BioRad CFX Connect Real-Time system was used with the following programme: 3 min of denaturation at 95 °C, followed by 39 cycles of 15 s at 95 °C, 30 s at 60 °C, and 30 s at 72 °C.

## Library preparation for community profiling

DNA extraction followed the same procedure described in the "Microbial Load Measurement" section, with the exception that separate extractions were conducted for bacteria and fungi. Specifically, 15 μl of the crushed suspension was split into two plates, where 0.2 pg of BI-124 spike was added per well for bacterial DNA extraction, while 0.046 ng of BI-124 spike was used for fungal extraction[52]. All other steps remained unchanged. The bacterial 16S rRNA gene (v5v7 variable regions) and the fungal ITS1 region were amplified in 96-well plates using PCR with primer pairs 16S-799F/16S-1192R and ITS1F/ITS2, respectively (PCR I; Supplementary Data 3). Each PCR I reaction (25 μl total volume) was prepared with 0.4 μl DFS-Taq polymerase (BIORON), 2.5 μl 10x incomplete buffer (BIORON), 0.5 μl of 10 mM MgCl$_2$, 2.5 μl of 3% BSA, 0.5 μl of 10 mM dNTPs, 0.75 μl of 10 μM forward primer (799 F for 16S or ITS1F for ITS), 0.75 μl of 10 μM reverse primer (1192 R for 16S or ITS2 for ITS), and 1 μl of extracted DNA, adjusted to 25 μl with nuclease-free water. The thermocycling conditions were: initial denaturation at 94 °C for 2 min, followed by 25 cycles of 94 °C for 30 seconds, 55 °C for 30 s, and 72 °C for 1 min, with a final extension at 72 °C for 10 minutes. To remove residual primers and dNTPs, each 25 μl PCR reaction was treated with 1 μl Antarctic phosphatase (New England BioLabs), 1 μl Exonuclease I (New England BioLabs), and 3 μl Antarctic phosphatase buffer, incubated at 37 °C for 30 minutes, and then heat-inactivated at 85 °C for 15 minutes. The digested PCR product was centrifuged at 2,000xg at 4 °C for 10 minutes. A second PCR was then conducted using barcoded primers containing Illumina adapters specific to each sample (PCR II; Supplementary Data 3). Each 25 μl PCR II reaction contained 0.4 μl DFS-Taq polymerase, 2.5 μl 10× incomplete buffer, 0.5 μl of 10 mM MgCl$_2$, 2.5 μl of 3% BSA, 0.5 μl of 10 mM dNTPs, 0.75 μl of 10 μM forward primer, 0.75 μl of 10 μM reverse primer, and 3 μl of the purified PCR I product, adjusted to 25 μl with nuclease-free water. The thermocycling conditions included an initial denaturation at 94 °C for 2 min, followed by 10 cycles of 94 °C for 30 s, 55 °C for 30 s, and 72 °C for 1 min, with a final extension at 72 °C for 10 min. To verify amplification success, 5 μl of each PCR product was mixed with 1 μl Orange DNA Loading Dye (Sigma-Aldrich), run on a 1% agarose gel

containing 0.05% EtBr, and electrophoresed at 110 mV. The expected product (~500 bp) contained the microbial sequences (v5v7 regions for bacteria or ITS for fungi), along with sample-specific barcodes and Illumina adapters. Successfully amplified products were purified using AMPure XP (Beckman Coulter) following the manufacturer's protocol. DNA concentrations were measured using the Quant-iT dsDNA Assay Kit (Invitrogen), and samples were pooled in equimolar proportions within full factorial replicates. The pooled libraries underwent two rounds of AMPure XP purification, and their final concentrations were assessed via qPCR using the KAPA Library Quantification Kit (Roche, Basel, Switzerland). Amplicon libraries were combined with 10% PhiX and sequenced using an in-house Illumina MiSeq platform with 500-cycle v2 reagent kits and custom sequencing primers (Supplementary Data 3).

## Microbial community analysis

Paired 16S rRNA amplicon sequencing reads were merged using join_paired_ends (QIIME2-2023-2, default settings), followed by quality filtering and demultiplexing via split_libraries_fastq (QIIME2-2023-2) with a maximum barcode error threshold of 1 and a minimum Phred score of 30. Unique amplicon sequence variants (ASVs) were identified from error-corrected reads and further processed to remove chimeric sequences using DADA2 (version 1.26.0). Taxonomic assignments were determined by mapping ASVs against a reference FASTA file containing sequences from our SynCom strains. ASVs not aligning with the reference database accounted for less than 0.001% of reads were therefore excluded. Count tables were generated based on the mapping results. To normalize microbial read counts, BI-124 spike-in[52] read counts were initially used and subsequently removed from further analyses. Beta-diversity was assessed using Bray-Curtis distances[53] and visualized via Principal Coordinate Analysis (PCoA). Differences in microbial composition across conditions were evaluated through canonical analysis of principal coordinates (capscale function, vegan package version 2.6-6.158[54]), followed by permutational multivariate analysis of variance (PERMANOVA[55]). Finally, heatmap was constructed using log-transformed absolute read abundances and plotted with ggplot2 (version 3.5.1).

## Phylogenetic trees of the SynCom

Phylogenetic trees of the MiniSynCom were constructed independently for bacteria and fungi. 16S rRNA gene and ITS1 fungal sequences were aligned with DECIPHER version 2.16.1[56], and neighbor-joining phylogenetic trees were constructed using Phangorn version 2.5.5. Trees were visualized with the online tool "interactive Tree Of Life" (iTOL version 6.9.1).

## Exometabolomic sample processing

For targeted and untargeted TCA-Glycolysis, sample processing was conducted using Dionex Integrion HPIC System (Thermo Fisher) anion exchange chromatography[57]. Separation was performed with a Dionex Ionpac AS11-HC column (2 mm × 250 mm, 4 µm particle size, Thermo Fisher) at 30 °C. A guard column, Dionex Ionpac AG11-HC b (2 mm × 50 mm, 4 µm particle size, Thermo Fisher), was placed before the separation column. The eluent (KOH) was generated in situ by a KOH cartridge and deionized water. At a flow rate of 0.380 mL/min a gradient was used for the separation: 0–0.5 min 20 mM KOH, 0.5–7 min 20–30 mM, 7–9 min 30–90 mM, 9–11 min 90–100 mM, 11–13 min 100 mM, 13.01–15 min 20 mM. A Dionex suppressor AERS 500, 2 mm was used for the exchange of the KOH and operated with 95 mA at 17 °C. The suppressor pump flow was set at 0.6 mL/min. Samples were re-dissolved in 100 µL of MilliQ and injected from an 8 °C temperated autosample using full loop mode (5 µL). The Integrion was connected to QExactive mass spectrometer (Thermo Fisher). The Q-Exactive was operated in negative ionization mode, scanning either in Full-MS mode within a mass range of m/z 50 to 750 (Fig. 2) or in Full-MS/ddMS2

(Top10) mode within a mass range of m/z 70 to 900 (Figs. 3 and 4). For ddMS2, the resolution was set at 17,500. The maximal injection time was set to 200 ms, and the HESI source was operating with a spray voltage of 2.75 kV. The ion tube transfer capillary temperature was 350 °C, the sheath gas flow 50 arbitrary units (AU), the auxiliary gas flow 14 AU, and the sweep gas flow was set to 3 AU at 380 °C. All samples were analyzed in a randomized run order. We provide support for level 1 identification[58] for TCA-Glycolysis metabolites, by confirming the structure by matching the main fragmentation of the reference standard using MS/MS and the retention time (Supplementary Data 4-7).

For Targeted AminoAcids, 40 µL of the re-dissolved (in water) samples were derivatized adding in order 20 µL of 100 mM Sodium Carbonate + 20 µL 2% (MeCN) Benzylchloride and vortexed for 30 s. Two microlitres of the derivatized sample were injected onto a 100 × 1.0 mm HSS T3 C18 UPLC column, packed with 1.8 µm particles (Waters). The flow rate of the UPLC was set to 0.1 ml/min, and the buffer system consisted of buffer A (10 mM ammonium Formate and 0.15% Formic acid in UPLC-grade water) and buffer B (UPLC-grade acetonitrile). The UPLC gradient was as follows: 0–4.1 min 100–85% A, 4.1–4.5 min 85–83% A, 4.5–11 min 83–45% A, 11–11.5 min 45–30% A, 11.5–13 min 30–0% A, 13–14 min 0% A, 14–14.1 min 0–100% A, 14.1–20 min 100% A. This leads to a total runtime of 20 min per sample. The UPLC was connected to an Orbitrap, equipped with a heated ESI (HESI) source (QExactive, Thermo Fisher Scientific). The QExactive mass spectrometer was operating in positive ionization mode scanning Full-MS in a mass range between m/z 70 and 900. The maximal injection time was set to 200 ms, and the HESI source was operating with a spray voltage of 4.00 kV. The ion tube transfer capillary temperature was 350 °C, the sheath gas flow 50 arbitrary units (AU), the auxiliary gas flow 35 AU, and the sweep gas flow was set to 1 AU at 380 °C. All samples were analyzed in a randomized run order.

## Exometabolomics data analysis

For targeted TCA-Glycolysis and amino acids, data analysis was performed using the Quan module of the Trace Finder 4.1 software (Thermo Fisher Scientific) in combination with a sample-specific compound database, derived from measurements of commercial reference compounds. Targeted metabolites consist of 16 TCA-glycolysis metabolites and 19 amino acids (Supplementary Table 3). These amino acids and metabolites were selected because they play critical roles in fundamental microbial processes, such as protein synthesis, energy production, stress response, and metabolic regulation. They encompass essential amino acids, intermediates of central metabolic pathways (e.g., TCA cycle, glycolysis), and key carbon sources, providing a comprehensive representation of microbial metabolism and allowing for a broad analysis of metabolic shifts in response to experimental conditions.

For untargeted metabolite data analysis, the raw data files were converted to open mzML formats and analyzed with MZmine 3.0[59]. The MZmine batch file contained the complete configuration of a list of processing steps, modules and related parameters for the processed samples. The output files produced by MZmine were exported for further data analysis. MS spectra and list of features were stored in.csv and. mgf. Metadata file listed the input of raw data files and corresponding sample information based on the experimental design of the study.

All the plots and statistics were performed on R. Targeted data are TIC-normalized (to avoid technical variation due to the period of acquisition of the scans for targeted data) and log-transformed. For untargeted data, we subtracted the mean peak area of the blank samples (n = 4) from all sample values and normalized the data by calculating the relative abundance of each metabolite within each sample. Metabolite profiles were ordinated by PCA and significance was determined by a permutational analysis of variance test (Adonis2 function) between

conditions (permutations = 999). For display purposes, log values of 0 were expressed as the minimal value detected in the analysis.

## Plant growth assay

Exometabolites collected throughout the MetaFlowTrain experiment were pooled from all replicates of each condition (n = 4) and filtered through 0.22 μm filters. Forty-five milliliters of the pooled exometabolites and 5 mL of 10% agarose were combined in a Falcon tube, thoroughly mixed, and poured into square plates. Each plate was seeded with two rows of 15 sterilized and stratified seeds, totaling 30 technical replicates per plate (refer to the "Plant Growth Conditions" section for details). Two plates, corresponding to two distinct biological replicates, were used per condition. Germination rates were calculated as the ratio of germinated seeds to the total seeds planted (n = 60), and root lengths were measured using ImageJ (v1.53a).

## MetaFlowTrain experiment connected to the hydroponic system

For this experiment, we equipped an axenic hydroponic system consisting of black-walled 50-mL Falcon tubes with 3D-printed collar (Supplementary Data 1-2[24]) enclosing a wet peat sponge. Two platinum-cured silicone tubes (15 m, SE-TUB-SIL-1*1, Darwin Microfluidics) were inserted into the bottom of each Falcon tube, passing through the peat sponge. The Falcon tubes were mounted on a rack inside sterile boxes (TP5000 + TPD5000, Sac O2), which featured a hole on each side to allow silicon tubes to pass through. These holes were sealed with autoclavable polyurethane foam.

For the plant conditions, sterile 2-week-old *Arabidopsis* seedlings, grown on ½ MS medium with 1% sucrose (Murashige & Skoog Medium with Vitamins, Duchefa), were transferred into the peat sponges, which had been pre-cut in half. The roots penetrated the sponge and grew freely into the medium below. The non-planted condition consisted of the hydroponic system without plants. The planted and non-planted hydroponic systems were then incubated for four weeks at 21 °C, with a light cycle of 10 h at 19 °C and 14 h of darkness. Each box contained one planted and one non-planted hydroponic system, and each box represented one replicate.

After four weeks, the hydroponic systems were serially connected to the MetaFlowTrain via the silicone tubes. The setup included an initial pump delivering ½ MS medium from a media bottle to the hydroponic system, which was then connected to a second pump that supplied the microbial microchambers. Tubing from the hydroponic systems passed through the box holes, with one tube connecting the media bottle to the first pump, and the other linking the Falcon tube to the second pump, which fed into the microbial microchambers. The holes were kept sterile using the autoclaved foam inserts. Both pumps were operated at a flow rate of 7.349 μL/min, consistent with other MetaFlowTrain experiments.

Microchambers were inoculated with the multikingdom SynCom (BF) as previously described. We ran the MetaFlowTrain experiments for 6 days, then harvested the microbes from the microchambers and collected metabolites as previously described.

## Statistics and reproducibility

R version 4.2.2 was used to perform all analyses and vizualisation. No statistical method was used to predetermine sample size. Amplicon sequencing data were analyzed using QIIME2 (v2023.2) and DADA2 (v1.26.0). Sequence alignments were constructed using DECIPHER (v2.16.1), and phylogenetic trees were built using Phangorn (v2.5.5). Visualization of the trees was performed using the online tool iTOL (v6.9.1). Exometabolite-targeted samples were analyzed with Trace Finder 4.1 software (Thermo Fisher Scientific), while untargeted metabolite data analyses were conducted using MZmine (v3.0). All visualizations, except for the trees, were created using ggplot2 (v3.5.1). Non-parametric tests were performed using the Kruskal-Wallis test,

followed by two-sided Dunn's post-hoc test with Benjamini-Hochberg (BH) adjustment for multiple comparisons, implemented using the PMCMRplus (v1.9.10), dunn.test (v1.3.6), and rcompanion (v2.4.36) R packages. PERMANOVA (PERmutational Multivariate ANalysis Of VAriance using distance matrices) was performed with the adonis2() function in the vegan package (v2.6-6.1), using Bray-Curtis distances for community profiling data and Canberra distances for metabolomics data. Significance was determined at P ≤ 0.05 and indicated by significance groups. For comparisons between two conditions, the non-parametric two-sided Wilcoxon test was applied using the wilcox.test function from the stats package (v4.2.2). Statistical significance between proportions was assessed using Fisher's exact test via the fisher.test function from the stats package (v4.2.2). The specific statistical tests used are indicated in each figure legend. Plant length measurements were performed using Fiji, and figures were assembled in Adobe Illustrator. No data were excluded from the analyses. All experiments involving plants grown on agar plates or in Flowpots were randomly distributed within the growth chambers and shuffled every week to minimize localization impact. For the MetaFlowTrain experiments, conditions were randomly distributed across media and pumps, with each replicate using a unique microchamber or microchamber train. During sample harvesting and processing, numeric identifiers were used instead of full sample names for simplicity and blinding.

## Reporting summary

Further information on research design is available in the Nature Portfolio Reporting Summary linked to this article.

## Data availability

The raw sequencing data from SynCom reconstitution experiments have been deposited in the European Nucleotide Archive (ENA) at EMBL-EBI under accession number PRJEB80329. The MS raw data of the targeted and untargeted metabolomics are deposited on MassIVE at ftp://massive.ucsd.edu/v06/MSV000096804/ (https://doi.org/10.25345/C5V40KB3M). We provide a protocol in PDF format (Supplementary Protocol 1), with the latest version accessible on protocols.io (https://doi.org/10.17504/protocols.io.36wgqd68ovk5/v3)[60]. The complete set of 3D printing models and stereolithography files can be downloaded via Zenedo (https://doi.org/10.5281/zenodo.15020528)[24]. Future updates to the 3D files will be available in the online protocol in the Material section. Datasets can be accessed via GitHub (see Code Availability) or through the source data provided with this paper. Source data are provided with this paper.

## Code availability

All scripts and data sets employed in this work are available from GitHub at https://github.com/gchesneau53/Script_from_Chesneau_et_al_2025 [61].

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

## Acknowledgements

This work was supported by funds to S.H. from a European Research Council consolidator grant (MICROBIOSIS 101089198), the Cluster of Excellence on Plant Sciences (CEPLAS) and the Priority Programme: Deconstruction and Reconstruction of the Plant Microbiota (SPP DECRyPT 2125; HA 8169/2-2), both funded by the Deutsche Forschungsgemeinschaft. We would like to thank Mehdi Moussaoui for creating and editing the Supplementary Movie 1, and Aude Geistodt-Kiener for providing the *Chlamydomonas reinhardtii CC1690* (*Cr*) strain and the associated protocols. Thanks to Sebastian S. Cocioba for sharing his insights regarding the hydroponic system and Neysan Donnelly for editing the manuscript.

## Author contributions

G.C. and S.H. conceptualized the research and experimental design. G.C. conducted the experiments and designed 3D-printed parts. J.H. established the hydroponic system and participated in the experimental design, execution, and analysis of the connected hydroponic and MetaFlowTrain experiment. S.P. performed the mass spectrophotometry and data processing. G.C. and S.W. developed the protocol. G.C. and S.H. created the figures and wrote the manuscript. All co-authors contributed to editing the paper.

## Funding

## Competing interests

The authors declare no competing interests.
