## [Transparent Peer Review file · Nature Communications]

MetaFlowTrain: a highly parallelized and modular fluidic system for studying exometabolite-mediated inter-organismal interactions

Corresponding Author: Dr Stéphane Hacquard

Version 0:

Reviewer comments:

Reviewer #1

(Remarks to the Author)

Hacquard and colleagues present a novel experimental setup, the MetaFlowTrain, that allows metabolite exchange between microorganisms. The system is based on microchambers that are 3D-printed, can be used sterile with different growth media. 24 chambers can be assembled and run in parallel, and up to 6 chambers can be connected in one series to allow metabolite exchange. The system presented meets a need in many biological fields studying the (metabolic) interactions between organisms. The authors tested the suitability of the system in answering multiple different research questions from effects of pre-conditioned substrates on bacterial and fungal growth, to inhibitory effects of microbial metabolites on microbe growth. They show that their system supports the growth of a variety of small organisms, and microbes can even be fed exudates of a plant attached to the system. This manuscript is timely and will meet considerable interest in the community.

I am outlining a few detailed comments and suggestions below:

Abstract: the authors describe well the functionality of the system. I suggest to add 1-2 sentences about the biological findings created with the system.

L120: when you sample from the chamber directly, what volume can you remove without disturbing the cells in the bottom? is a single sampling concentrated enough for metabolomics? Please expand on this in the text.

L155: please provide more detail on how much volume was collected for analysis, and for how long the metabolites were collected (the entire 62 h)? What is the minimum amount of peat extract volume needed for metabolomics & sequencing, and what was the maximum amount you tested? Did you concentrate the metabolite samples before analysis?

Figure 1c: to me it looks like in the small figure displaying the number of chambers a maximum of 5 chambers is shown? I suggest either adding a row.

Supplementary Protocol: check for typos (e.g. in figure legends)

Figure 3 and others: please add number of metabolites that were used to make the PCA plots. Why were TCA glycolysis metabolites and amino acids chosen for the plot in 3e? Are these the ones changing most? or are these all the metabolites changing? please elaborate.

Figure 4b: the figure would be more intuitive for me to read if the bacterial/fungal bars are arranged differently in order of how the trains were arranged. either above each other as now with the green ones flipped, or side-by-side as a stacked plot. please switch 4c and 4d in the figure so 4c is to the left.

Supplemental table: 'price per resin' is an empty sheet?

Please add supplemental tables for the metabolomics and microbiome data, at least for the targeted analysis, so one can see compound identity, peak height/area, number of compounds...

Reviewer #2

(Remarks to the Author)

Chesneau et al describe a very interesting and powerful fluidic system and its use for studying inter-organismal exometabolite-mediated interactions. As requested, I have primarily focused on the metabolomics aspects of the paper. That said, I find this to be a very exciting technical advance and one that I anticipate will be widely cited and adopted, especially given the detailed designs, videos, and protocols that accompany the manuscript.

General comments:

How do the authors distinguish between communication (title) and cross-feeding? Do they consider inhibition 'communication'. I think it may be more accurate to end the title with 'inter-organismal interactions'.

The link between the experiments in figures 4 and 5 is a bit weak. The authors should mention that *Pseudomonas* R401 is a member of the SynCom and provide a rationale for expecting that this inhibition is a result of pyoverdine, diacetylphloroglucinol, and brassicaeptin produced from this strain (e.g. based on its abundance and previous reports).

The paper discusses metabolic fluxes by changing the dilution rates but didn't really take advantage of this capability (e.g. comparing community responses and exometabolite abundances to across dilution rates). L358-361 asserts that this enables capability enables differentiating consumption vs. production--it would be good to add more explanation for this.

How did they decide on the media/extract concentration, dilution rates (7 uL/min), and collection time used in these experiments? I assume it took several preliminary experiments to determine these values such that they are sufficient for the exometabolomics and detectable community composition. I think it would be helpful to provide some guidance on how you selected these parameters.

Did the authors have issues with filter fouling? Any reason why this couldn't be run anaerobically to study gut microbes inside a glove box?

The SI videos and figures are very valuable in understanding how this system works--very nice! I assume that they will be linked in a way that is easy for the reader to access them (they weren't for me). If not, the authors should make a .pdf with the figure captions and figures.

I didn't see the details for how Cf was calculated in Figure 5. I'd expect the larger value to be the higher load but it seems like it's the opposite—that the Ct increases from 17 to 20+ with the R401 vs. the mutant.

Metabolomics:

It's great that the authors deposited the data as a MASSIVE project. Though, I'm a bit confused. The methods say "The QExactive was operating in negative ionization mode scanning Full-MS/ddMS2(TOP10) in a mass range between m/z 70 and 900. Resolution of the ddMS2 was settled at 17,500." yet it appears that there isn't any MS/MS in any of the deposited files despite having reasonable signal and apparently doing to 10 ddMS2 using a Q-Exactive tandem mass spectrometer: Hopefully this will open for you (the ms2 column = 0)

https://explorer.gnps2.org/MSV000095811/raw/230530_Guillaume_TCA&Gly/mzML/10_F_B_SA_R16.mzML?dataset_accession=MSV000095811&metadata_source=MASSIVE&metadata_option=

I see this more as a missed opportunity than a major issue, though. The MS/MS could be very helpful in looking for the metabolites that they speculate are inhibiting growth etc and could fit in nicely to support the rationale for the KO experiment. It would certainly strengthen the manuscript if they had evidence for production of these metabolites.

I see that there are some .csv and solid works files mixed in with the mass spec files. Though, I did not find the tables used to make the PCA plots and figure 3e. These and the other tables used to make figures should be provided in the SI (more detail below).

SI Table 4 has a list of amino acids and TCA-glycolysis metabolites that were used for the target analysis. However, I don't see any support for the Level 1 identification of these compounds according to community standards. The MS/MS would enable confirmation with the current community standards for level 1 identifications based on MS/MS, RT, and exact mass vs. standards (<https://pubs.acs.org/doi/full/10.1021/es5002105>). Given that apparently MS/MS was not collected and they are not claiming anything unexpected, I think it would be acceptable to state that metabolites were IDed on m/z RT and named according to an earlier standard (<https://doi.org/10.1007/s11306-007-0082-2>) provided that this is clear in the manuscript. They will have to provide some explanation why they didn't do MS/MS. If there is still some of these samples it would be much better to do some MS/MS on a few representative samples to provide rigorous level 1 identifications.

A SI table with the targeted metabolites identified comparing the retention times and accurate mass with the standards along

with the table(s) of ion abundance (peak height or area) across the samples in a given experiment should be provided (as mentioned above).

I assume that there were many compounds coming from the tubing? The water02.raw file certainly suggests this e.g. m/z 201.043. If so, the authors should mention this because this will likely impact the untargeted analysis and possibly change responses over the experiments due to matrix effects.

Trent Northen

Version 1:

Reviewer comments:

Reviewer #1

(Remarks to the Author)

my comments have been addressed, thank you.

Reviewer #2

(Remarks to the Author)

The author's have addressed my concerns and the manuscript is now suitable for publication.

REVIEWER COMMENTS

Reviewer #1 (Remarks to the Author):

Hacquard and colleagues present a novel experimental setup, the **MetaFlowTrain**, that allows metabolite exchange between microorganisms. The system is based on microchambers that are 3D-printed, can be used sterile with different growth media. 24 chambers can be assembled and run in parallel, and up to 6 chambers can be connected in one series to allow metabolite exchange. The system presented meets a need in many biological fields studying the (metabolic) interactions between organisms. The authors tested the suitability of the system in answering multiple different research questions from effects of pre-conditioned substrates on bacterial and fungal growth, to inhibitory effects of microbial metabolites on microbe growth. They show that their system supports the growth of a variety of small organisms, and microbes can even be fed exudates of a plant attached to the system. This manuscript is timely and will meet considerable interest in the community.

I am outlining a few detailed comments and suggestions below:

Abstract: the authors describe well the functionality of the system. I suggest to add 1-2 sentences about the biological findings created with the system.

We agree with the reviewer's suggestion regarding the omission of results in the abstract and have added a sentence (L25-27) to describe the biological findings: *"Using MetaFlowTrain, we uncover soil conditioning effects on synthetic community structure and plant growth, while confirming the potent inhibitory activity of bacterial exometabolites against fungi."*

L120: when you sample from the chamber directly, what volume can you remove without disturbing the cells in the bottom? is a single sampling concentrated enough for metabolomics? Please expand on this in the text.

Thank you for your comment. We do not disturb the cells in the chamber during sampling for metabolomics. We sample the cells within the chamber and the exometabolites from the final 15 mL Falcon tubes (as described in the section "MetaFlowTrain harvesting," L566-582). This method allows for precise sampling without disrupting the cells at the bottom of the chamber. We have added precision L120 now L124-127, to help the reader better understand the process: *"Multiple temporal outputs can be collected. Microbial cells are harvested through the hole at the top of each microchamber, while metabolite collection is performed from the 15 mL Falcon tubes at the end of each metabolic train (Fig. 1d)."*

In terms of metabolomics, the 2 mL volume used for analysis was sufficiently concentrated to provide reliable data for metabolic profiling in our study. Before conducting the experiments, we performed tests to ensure that this concentration, when used with our QExactive mass spectrometer, was optimal for detecting the targeted metabolites. We will expand on this in the manuscript to provide further clarification of our approach. It is important to note that the appropriate sample volume may vary depending on the specific study and the metabolites being targeted. To address this, we have added the following statement in lines 568-573: *"In our study, a*

2 mL concentrated sample was adequate for targeting metabolites from the TCA cycle, glycolysis, and amino acid metabolism. However, depending on the study's objectives, this volume may need to be adjusted, as it can vary based on factors such as the initial abundance of target metabolites, sample complexity, media composition, and instrument sensitivity, among others."

L155: please provide more detail on how much volume was collected for analysis, and for how long the metabolites were collected (the entire 62 h)? What is the minimum amount of peat extract volume needed for metabolomics & sequencing, and what was the maximum amount you tested? Did you concentrate the metabolite samples before analysis?

Thank you for your comment. We have added the following note to guide readers to this information (L155, now L160-162): "Please refer to the Methods section for detailed information on our approach to harvesting both microbes and exometabolites." The entire harvesting process is explained in the 'MetaFlowTrain harvesting' section of the Methods. In this section, we provide a comprehensive description of the sampling procedure, including the volume collected, the timing of metabolite collection, and the sample concentration.

Regarding the minimum and maximum amounts of peat extract needed for metabolomics and sequencing, we pre-tested the direct injection of our peat extract and a 2 mL concentrated peat extract. The 2 mL concentrated peat extract did not produce background interference on our spectra and allowed us to detect all targeted metabolites, so we selected this concentration for our study. For further details on the quality of targeted metabolite detection, please refer to the TraceFinder table uploaded on MASSIVE, which is now also available as Supplementary Tables 4-7, as well as the "level 1 identification" table provided on MASSIVE (<ftp://massive.ucsd.edu/v06/MSV000096804/>, L829).

Overall, the MetaFlowTrain system enables the sampling of very small amounts of exometabolites, up to tens of milliliters. Depending on experimental requirements, the samples can be concentrated or diluted as needed.

Figure 1c: to me it looks like in the small figure displaying the number of chambers a maximum of 5 chambers is shown? I suggest either adding a row.

Thank you for pointing this out. We have added an additional row to Figure 1c to better illustrate the chamber setup

Supplementary Protocol: check for typos (e.g. in figure legends)

We have carefully checked the Supplementary Protocol for typos, including in the figure legends. The changes have been made in Supplementary Protocol 1, and we have also created a fork with corrections on protocol.io.

Figure 3 and others: please add number of metabolites that were used to make the PCA plots. Why were TCA glycolysis metabolites and amino acids chosen for the plot in 3e? Are these the ones changing most? or are these all the metabolites changing? please elaborate.

Thank you for your comment. We have now included the number of targeted metabolites in the figure legends (n=35 metabolites) L1021-1022, 1034 and 1055-1056. Additionally, we have added the following explanation in lines 756-762 regarding the rationale for targeting these metabolites: “*These amino acids and metabolites were selected because they play critical roles in fundamental microbial processes, such as protein synthesis, energy production, stress response, and metabolic regulation. They encompass essential amino acids, intermediates of central metabolic pathways (e.g., TCA cycle, glycolysis), and key carbon sources, providing a comprehensive representation of microbial metabolism and allowing for a broad analysis of metabolic shifts in response to experimental conditions.*” This selection was made prior to the experiment, and we did not specifically choose metabolites based on whether they were changing the most in our experiments.

Figure 4b: the figure would be more intuitive for me to read if the bacterial/fungal bars are arranged differently in order of how the trains were arranged. either above each other as now with the green ones flipped, or side-by-side as a stacked plot. please switch 4c and 4d in the figure so 4c is to the left.

We have carefully considered your feedback regarding the arrangement of Figure 4b and have decided to keep it as is. We’ve tried different layouts and believe this layout, where the bars match with 4c, provides a clearer representation of the data and maintains consistency throughout the figure. Additionally, we have switched Figures 4c and 4d, as requested, so that 4c now appears to the left.

Supplemental table: 'price per resin' is an empty sheet?

Thank you for pointing this out. It appears that the file (Supplementary Table S2) became corrupted during the upload process because we did not remove all the unnecessary tabs. The issue has been addressed.

Please add supplemental tables for the metabolomics and microbiome data, at least for the targeted analysis, so one can see compound identity, peak height/area, number of compounds...

We provide all raw data with our manuscript. For microbiome data, please refer to the github in code availability, and for metabolomics data, refer to the MASSIVE repository, as detailed in the “Data availability” section (L822–834). All tables used to create plots are available on GitHub, as outlined in the “Code availability” section: “All scripts and datasets employed in this work are available from GitHub at https://github.com/gchesneau53/Script_Chesneau_et_al_2025” (L836–839).

Additionally, we now include Supplementary Tables 4, 5, 6, and 7, which provide metabolomics data for Level 1 identification. These tables include retention time (Actual RT), accurate mass measurements (m/z [Apex] and m/z [Delta]), ion abundance (Area), and height (Height) for all samples, pools, and standards. We also provide “Supplementary table 8” listing all the metabolites targeted in our study.

Reviewer #2 (Remarks to the Author):

Chesneau et al describe a very interesting and powerful fluidic system and its use for studying inter-organismal exometabolite-mediated interactions. As requested, I have primarily focused on the metabolomics aspects of the paper. That said, I find this to be a very exciting technical advance and one that I anticipate will be widely cited and adopted, especially given the detailed designs, videos, and protocols that accompany the manuscript.

General comments:

How do the authors distinguish between communication (title) and cross-feeding? Do they consider inhibition ‘communication’. I think it may be more accurate to end the title with ‘inter-organismal interactions’.

Thank you for your comment. We agree with your suggestion and have revised the title to: “*MetaFlowTrain: A highly parallelized and modular fluidic system for studying exometabolite-mediated inter-organismal interactions*”. L1-2.

The link between the experiments in figures 4 and 5 is a bit weak. The authors should mention that *Pseudomonas* R401 is a member of the SynCom and provide a rationale for expecting that this inhibition is a result of pyoverdine, diacetylphloroglucinol, and brassicaeptin produced from this strain (e.g. based on its abundance and previous reports).

Thank you for this comment. We would like to clarify that *Pseudomonas* R401 is not part of the SynCom used in our experiments (see Supplementary Table S1). The inhibition observed in Figure 4 was identified using our SynCom. Subsequently, we utilized the well-characterized *Pseudomonas* R401 strain and its triple mutant, which is depleted in antimicrobial exometabolites, as described in a previously published study (<https://doi.org/10.1038/s41467-024-48517-5>). This study demonstrated that pyoverdine, diacetylphloroglucinol, and brassicaeptin play key roles in inhibiting root-associated bacteria and fungi. Our goal was to provide proof of concept for the system's ability to use bacterial mutants depleted in exometabolite production to assess the exometabolic impact on other microbial cells using MetaFlowTrain. However, we cannot directly link the two experiments or claim that the three exometabolites solely explain the inhibition observed in Figure 4.

The paper discusses metabolic fluxes by changing the dilution rates but didn't really take advantage of this capability (e.g. comparing community responses and exometabolite abundances to across dilution rates). L358-361 asserts that this

enables capability enables differentiating consumption vs. production--it would be good to add more explanation for this.

You are correct that we did not explore the different flow rate capabilities of the system in our experiments but just tested its ability to support various flow rates (Supplementary Figure S2). Consequently, we have revised the discussion part you mentioned to not emphasize flow rate speed.

See lines 364-368: *“By controlling the direction of metabolic fluxes within the chamber(s), the system enables the detection of directional effects. Furthermore, while medium consumption still occurs within the chambers, the continuous influx driven by the peristaltic pump significantly reduces the confounding effects of medium consumption versus metabolite production, which are common in closed systems.”*

How did they decide on the media/extract concentration, dilution rates (7 uL/min), and collection time used in these experiments? I assume it took several preliminary experiments to determine these values such that they are sufficient for the exometabolomics and detectable community composition. I think it would be helpful to provide some guidance on how you selected these parameters.

For all soil extracts we used the FlowPots system (Kremer et al., 2021), which is a cut syringe where we can grow plants. Since it is a syringe we can flush the whole pot with water to extract the soil metabolites. We flushed all the Flowpots with 20mL since this is the volume of the pot and combined all the flushes per condition. We developed this strategy for this study, the rationale of the extract concentration is based on a volume/volume ratio. We explain this strategy in the method section, and added this sentence to clarify our volume strategy *“We selected 20 mL as it corresponds to the volume of the Flowpot.”* L465-466.

For the dilution rate of 7 uL/min. We wanted to use the technically lowest possible flow rate to enhance metabolic impact between microbes in the different chambers. It appears that our pumps with the specific diameters of our hoses are suitable for a minimal flow rate speed of 7 uL/min. We added this explanation in the method section to clarify the use of this specific speed *“We utilize a flow rate of 7.3 μ L/min, since this is the lowest possible flow rate compatible with our pump and hose dimensions. The aim is to maximize contact time and, consequently, enhance the impact of metabolites on microbial interactions between the microchambers.”* L560-563.

For the collection time, we were more limited by the initial media input, since we used natural soil extracts for most of our study. Preparing large amount of the soil extracts using the flowpots system is quite a lot of work but the overall amount of media is a limiting factor for the duration of the experiment. Using artificial media increases thoroughly the potential timescale of the experiments, since input media is not a limitation. We defined 62 hours since this was the time point showing most of the differences for metabolomics data and also significant growth of our strains. We clarified the rationale of the sampling time points in the text : *“We chose a 24-hour sampling time to capture an early snapshot of the metabolomic profile, while 62 hours was selected as it allowed for observing both bacterial and fungal growth. Although*

the experiment could have been extended, the use of peat extract, a natural rather than artificial medium, limited the volume we could produce.” L574-578.

We indeed, performed several first trial to set all this up. However these parameters are largely dependent on the strains used, the media, and the purpose of the study. We provide details about all of this in our online protocol on protocol.io.

Did the authors have issues with filter fouling? Any reason why this couldn't be run anaerobically to study gut microbes inside a glove box?

We have never observed filter fouling during the timeline of our experiments under the current experimental conditions. This is shown in Supplementary Figure S4c, which demonstrates the absence of flow rate reduction over 62 hours with our BF SynCom (lines L478–480). Addressing potential filter clogging was a major concern during the initial design of the system, particularly when working with fungal strains. To mitigate this, we developed a specific chamber design featuring an enlarged bottom section with a toothed surface. This, combined with the slow flow rate, effectively prevents filter clogging.

Additionally, we see no restrictions on using the system under anaerobic conditions, as its compact size allows it to fit within standard anaerobic chambers. In fact, we believe the system holds great potential for studying interaction between gut microbiota members. We have included a statement in the “Discussion” highlighting its potential applicability in anaerobic conditions. *“Although not tested in this study, the MetaFlowTrain system is expected to function under anaerobic conditions, as its compact design is compatible with standard anaerobic chambers.” L389-392.*

The SI videos and figures are very valuable in understanding how this system works—very nice! I assume that they will be linked in a way that is easy for the reader to access them (they weren't for me). If not, the authors should make a .pdf with the figure captions and figures.

Thank you for your positive feedback! We initially made the videos private for patent purposes, but they are now fully available online. We will ensure the videos and figures are properly linked for easy access to readers. Please note that we have already included supplementary Figure S1 with captions for the videos for further convenience.

I didn't see the details for how Cf was calculated in Figure 5. I'd expect the larger value to be the higher load but it seems like it's the opposite—that the Ct increases from 17 to 20+ with the R401 vs. the mutant.

Thank you for your comment. The Ct (cycle threshold) value in qPCR represents the cycle number at which the fluorescence signal exceeds a set threshold, indicating the presence of the target DNA. A lower Ct value means a higher amount of DNA in the sample, as fewer cycles are needed to detect the target, while a higher Ct value indicates a lower DNA load, requiring more cycles to reach the threshold.

Metabolomics:

It's great that the authors deposited the data as a MASSIVE project. Though, I'm a bit confused. The methods say “The QExactive was operating in negative ionization mode

scanning Full-MS/ddMS2(TOP10) in a mass range between m/z 70 and 900. Resolution of the ddMS2 was settled at 17,500.” yet it appears that there isn't any MS/MS in any of the deposited files despite having reasonable signal and apparently doing to 10 ddMS2 using a Q-Exactive tandem mass spectrometer: Hopefully this will open for you (the ms2 column = 0)
https://explorer.gnps2.org/MSV000095811/raw/230530_Guillaume_TCA&Gly/mzML/10_F_B_SA_R16.mzML?dataset_accession=MSV000095811&metadata_source=MASSIVE&metadata_option=

It is true that the content of the MASSIVE file storage was not clearly organized, and the methods section did not adequately describe our datasets and procedures. We appreciate this observation.

To clarify we have two datasets:

- 1- Dataset “230530” (Figures 2): This dataset was measured in Full-MS mode only, with no MS2 acquisition.
- 2- Dataset “221208” (Figure 3-4): This dataset was measured in Full-MS/ddMS2 (Top10) mode, as described in the methods section.

To address these concerns, we have revised the methods section to clarify the data acquisition settings: " *The Q-Exactive was operated in negative ionization mode, scanning either in Full-MS mode within a mass range of m/z 50 to 750 (Figure 2) or in Full-MS/ddMS2 (Top10) mode within a mass range of m/z 70 to 900 (Figure 3 and 4). For ddMS2, the resolution was set at 17,500*" (Lines 720–723)

Additionally, we have created a new MASSIVE project to include supplemental information that provides a detailed description of the datasets. (<ftp://massive.ucsd.edu/v06/MSV000096804/>, L829)

We hope these clarifications and updates address the concerns raised. Thank you for bringing this to our attention.

I see this more as a missed opportunity than a major issue, though. The MS/MS could be very helpful in looking for the metabolites that they speculate are inhibiting growth etc and could fit in nicely to support the rationale for the KO experiment. It would certainly strengthen the manuscript if they had evidence for production of these metabolites.

We agree that an untargeted analysis to identify metabolites potentially involved in growth inhibition (Figure 4) could provide valuable additional evidence and further support the rationale for the knockout (KO) experiment (Figure 5).

However, the primary focus of this paper is to describe the method and demonstrate proof of concept for the system. While metabolite identification is one of our broader research goals, we did not want to further investigate MS/MS data for this particular experiment, as the scope of this work was limited to validating the system and its functionality.

We recognize the potential for future studies to build upon our findings by including to explore the identity and roles of these metabolites in greater detail.

I see that there are some .csv and solid works files mixed in with the mass spec files. Though, I did not find the tables used to make the PCA plots and figure 3e. These and the other tables used to make figures should be provided in the SI (more detail below).

We acknowledge that the MASSIVE project currently includes some .sld and .csv files, which are used to import sequences into the machine software but are not relevant for downstream data analysis. Thank you for highlighting this point.

To address this, we created a new MASSIVE archive (<ftp://massive.ucsd.edu/v06/MSV000096804/>, L829). We have removed all unnecessary .sld and .csv files. Our MASSIVE archive now includes only the raw data along with the export files from TraceFinder (TraceFinder files include: Retention time (Actual RT), accurate mass measurements (m/z (Apex)/m/z (Delta)), and ion abundance (Area) and Height (Height) for all samples, pools and standards. Information on the standards; Mix 40: A mixture of 40 standards at 250 ng/mL concentration; Mix AA: A mixture of 23 (non-)essential and secondary amino acids (AAs) at 500 ng/mL concentration). We have also provided more details in the MASSIVE archive description.

We provide all tables used to make plot in the GitHub provided in the section “Code availability”: “All scripts and data sets employed in this work are available from GitHub at https://github.com/gchesneau53/Script_Chesneau_et_al_2025” (L836-839)

SI Table 4 has a list of amino acids and TCA-glycolysis metabolites that were used for the target analysis. However, I don't see any support for the Level 1 identification of these compounds according to community standards. The MS/MS would enable confirmation with the current community standards for level 1 identifications based on MS/MS, RT, and exact mass vs. standards (<https://pubs.acs.org/doi/full/10.1021/es5002105>). Given that apparently MS/MS was not collected and they are not claiming anything unexpected, I think it would be acceptable to state that metabolites were IDed on m/z RT and named according to an earlier standard (<https://doi.org/10.1007/s11306-007-0082-2>) provided that this is clear in the manuscript. They will have to provide some explanation why they didn't do MS/MS. If there is still some of these samples it would be much better to do some MS/MS on a few representative samples to provide rigorous level 1 identifications.

A SI table with the targeted metabolites identified comparing the retention times and accurate mass with the standards along with the table(s) of ion abundance (peak height or area) across the samples in a given experiment should be provided (as mentioned above).

Thanks for your comment. For the level 1 identification, we have now uploaded the TraceFinder table in the new MASSIVE archive (<ftp://massive.ucsd.edu/v06/MSV000096804/>, L829) and we also provide them as supplementary tables (**Supplementary Table 4-7**). TraceFinder files include:

Retention time (Actual RT), accurate mass measurements (m/z (Apex)/ m/z (Delta)), and ion abundance (Area) and Height (Height) for all samples, pools and standards. From Trace Finder tables we have inferred the Delta m/z (Expected m/z – actual m/z) for the respective metabolites in all samples. The difference was less than 10ppm for all of them.

Additionally, we have included an Excel file in the MASSIVE archive (Level 1 identification) comparing the average retention times (RTs) for standards, pooled samples, and individual samples from the MS1 measurements. This file also includes the RT differences between standards and pooled samples, which are all within 0.1 min, providing robust Level 1 identifications. Moreover, the file contains the fragmentation patterns for TCA and glycolysis metabolites measured in the sample set dated 221208. These patterns were compared with and confirmed against the MoNA database spectra.

We have also updated the Methods section to include a note on Level 1 identification, referencing Schymanski et al., 2014. The added statement reads:

“We provide support for level 1 identification for TCA-Glycolysis metabolites, by confirming the structure by matching the main fragmentation of the reference standard using MS/MS and the retention time (Supplementary Table 4-7).” L727-730.

For the amino acid (AA) analysis, we measured exclusively in MS1. This approach is a well-established method for amino acid quantification (<https://doi.org/10.1016/j.chroma.2017.07.061>), and MS2 scans are typically not performed for this analysis. Additionally, an amino acid mix was included in both experiments, covering the most critical amino acids.

The decision not to measure MS2 for the 230530 dataset was made in an effort to optimize the MS1 acquisition. Our goal was to capture metabolites at lower concentrations, as detecting low-abundance molecules can be particularly challenging in samples with a media matrix that contains high concentrations of soluble salts.

I assume that there were many compounds coming from the tubing? The water02.raw file certainly suggests this e.g. m/z 201.043. If so, the authors should mention this because this will likely impact the untargeted analysis and possibly change responses over the experiments due to matrix effects.

Thank you for your valuable comment. You are correct that we did not normalize features based on our blank samples. To address this, we reanalyzed Figure 2c (untargeted) using two different blank normalization strategies:

Strategy 1: We removed the 1,975 features consistently detected across the 4 blank replicates and reran the analysis.

Strategy 2: We subtracted the mean blank peak area for each feature across all samples.

Both approaches produced consistent results, with no significant change in the overall variation between the metabolic profiles of the strains. The variance explained was

35.6% (P = 0.001) for Strategy 1, 37.2% (P = 0.001) for Strategy 2, and 37.4% (P = 0.001) for the initial analysis, which did not account for blank normalization.

We chose to implement Strategy 2 in the revised manuscript, as it provides a quantitative approach that more accurately reflects the true metabolomic profiles of the strains. We appreciate you raising this point, as it allowed us to further validate our findings and ensure robustness.

We have updated Figure 2, modified the result section accordingly (L170) and added the blank normalization approach to the Methods section (Lines 773–775):

“For untargeted data, we subtracted the mean peak area of the blank samples (n = 4) from all sample values and normalized the data by calculating the relative abundance of each metabolite within each sample.”

Trent Northen